# Exposing Attention Glitches with Flip-Flop Language Modeling

**Bingbin Liu**[1]   **Jordan T. Ash**[2]  **Surbhi Goel**[3]  **Akshay Krishnamurthy**[2]  **Cyril Zhang**[2]

[1]Carnegie Mellon University     [2]Microsoft Research NYC     [3]University of Pennsylvania

bingbinl@cs.cmu.edu, surbhig@cis.upenn.edu,
{ash.jordan, akshaykr, cyrilzhang}@microsoft.com

## Abstract

Why do large language models sometimes output factual inaccuracies and exhibit erroneous reasoning? The brittleness of these models, particularly when executing long chains of reasoning, currently seems to be an inevitable price to pay for their advanced capabilities of coherently synthesizing knowledge, pragmatics, and abstract thought. Towards making sense of this fundamentally unsolved problem, this work identifies and analyzes the phenomenon of *attention glitches*, in which the Transformer architecture's inductive biases intermittently fail to capture robust reasoning. To isolate the issue, we introduce *flip-flop language modeling* (FFLM), a parametric family of synthetic benchmarks designed to probe the extrapolative behavior of neural language models. This simple generative task requires a model to copy binary symbols over long-range dependencies, ignoring the tokens in between. We find that Transformer FFLMs suffer from a long tail of sporadic reasoning errors, some of which we can eliminate using various regularization techniques. Our preliminary mechanistic analyses show why the remaining errors may be very difficult to diagnose and resolve. We hypothesize that attention glitches account for (some of) the closed-domain hallucinations in natural LLMs.

## 1   Introduction

Recent advancements in scale have yielded large language models (LLMs) with extraordinary proficiency in nuanced reasoning with factual knowledge. Despite these achievements, LLMs are known to produce incorrect outputs, often referred to colloquially as "hallucinations" or "distractions" (Ji et al., 2023). Generally, hallucinations refer to the phenomenon that a model's outputs are syntactically and grammatically accurate but factually incorrect. There are various types of hallucinations, and the focus of this work is the "closed-domain" variety (Saparov and He, 2022; OpenAI, 2023), where the model predictions contain factually incorrect or made-up information *according to a given context*, regardless of their correctness in the real world.

Perhaps surprisingly, such hallucinations can be observed even on simple algorithmic reasoning tasks. As a warmup, consider the queries shown in Figure 1 (and Appendix B.1), where we prompt LLMs to solve addition problems of various lengths. The responses simultaneously illustrate the following:

1. *Nontrivial algorithmic generalization:* In cases where the models succeed, it is unlikely that these exact numerical sequences appeared in the training data. To correctly output the first digit of the answer, the LLM must resolve a long dependency chain which generally depends on every digit in the input. Somewhere within these networks' internal representations, implementations of addition algorithms have emerged.

37th Conference on Neural Information Processing Systems (NeurIPS 2023).

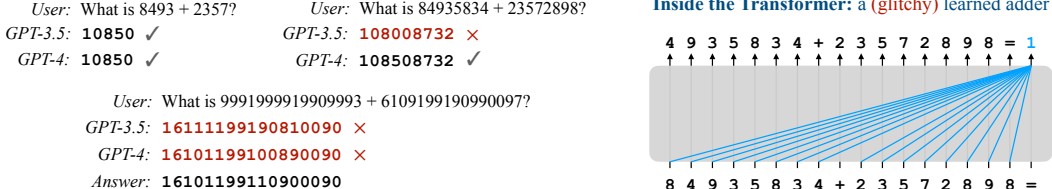

Figure 1: Cherry-picked integer addition prompts, showing how state-of-the-art LLMs can generalize non-trivially on algorithmic sequences, but sporadic reasoning errors persist. The first digit of the correct answer depends on every input; thus, an autoregressive model must propagate a "carry" bit across these long-range dependencies in a single pass. This (and many other algorithmic reasoning capabilities) can be implemented by a Transformer model using internal *flip-flops*.

2. *Sporadic errors ("hallucinations"):* These internal algorithms can be brittle and unreliable, especially when processing long inferential chains. Their failures can be subtle and unpredictable.

In this work, we consider the task of processing the *flip-flop language*, a minimal unit of sequential computation which consists of memory operations on a single bit (see Definition 1) and underlies virtually all[1] syntactic parsing and algorithmic reasoning capabilities (including implementing adders, and far more). A *flip-flop language modeling* (FFLM) task is defined on sequences of `write`, `read`, and `ignore` instructions: `write` sets the memory state to a certain value which is later retrieved by `read`, while `ignoring` any contents in between. We find that when trained to complete flip-flop sequences, the Transformer architecture exhibits a long tail of reasoning errors (incorrect `read` retrievals), unlike previous-generation recurrent models such as the LSTM (Hochreiter and Schmidhuber, 1997). We coin the term *attention glitch* for this phenomenon, and hypothesize that this captures a systematic failure mode of Transformer-based LLMs when internally manifesting long chains of algorithmic reasoning.

Our contributions are as follows:

- **FFLM: a minimalistic long-range dependency benchmark.** We propose *flip-flop language modeling*, a parametric family of synthetic benchmarks for autoregressive sequence modeling, designed to isolate and probe reasoning errors like those demonstrated in Figure 1. We view FFLM as a robust complement to the Long Range Arena (Tay et al., 2020) and some of the tests in BIG-Bench (Srivastava et al., 2022), and recommend measuring glitch rates as a **"stress test"** for architectural innovations in sequence modeling.[2]

- **Main empirical result: attention attends glitchily.** We find that while Transformer models can appear to learn flip-flop languages perfectly on held-out samples from the training distribution, they make a long tail of unpredictable reasoning errors (*attention glitches*), on both long-range and short-range dependencies. We evaluate various direct and indirect mitigations, including commonly-used regularization techniques and **attention-sharpening regularizers** — a plug-and-play way to sparsify self-attention architectures. We find that attention sharpening reduces reasoning errors by an order of magnitude, but none of our attempts were successful in driving the number of errors to exactly 0. Meanwhile, recurrent models work perfectly.[3]

- **Preliminary mechanistic analyses.** We provide some theoretical and empirical explorations which account for some of the internal mechanisms for attention glitches, and why they are so difficult to eliminate completely.

## 1.1 Related work

The challenge of learning long-range dependencies is a long-standing one in the statistical modeling of sequences (Samorodnitsky et al., 2007). The Transformer architecture (Vaswani et al., 2017), a paradigm-shifting sequence model, enables the scalable learning of a feedforward hierarchy of

---

[1]More precisely, whenever the desired algorithm needs to "store memory" (i.e. contains a non-invertible state transformation); see Section 3.2.

[2]To get started, see our data release: `https://huggingface.co/datasets/synthseq/flipflop`.

[3]It could be the case that recurrent models may fail at extremely long dependencies (Khandelwal et al., 2018).

meaningful long-range dependencies. Yet, factual errors over long contexts persist in these models; this is the subject of many careful studies in deep NLP (Khandelwal et al., 2018; Tay et al., 2020; Guo et al., 2022; Ji et al., 2023).

The sporadic non-factual outputs of LLMs have been popularly called "hallucinations", especially when there is an expectation that producing a correct answer is disproportionately "easy". Popular approaches for improving robustness to such errors include *chain-of-thought generation* (explicitly outputting intermediate reasoning steps) (Nye et al., 2021; Wei et al., 2022b) and enforcing self-consistency (Wang et al., 2022). In the emerging taxonomy of LLM pathologies (Saparov and He, 2022; Ji et al., 2023), the hallucinations studied in this work are of the *closed-domain* variety, under a deterministic notion of factuality (namely, consistency with flip-flop transitions) which is unambiguously reflected by the training data. We provide further discussion on the connections to natural LLM hallucinations in Section 6 and Appendix A.4.

**Long-range dependency and reasoning benchmarks.** Many datasets and benchmarks have been designed to isolate qualitative issues in langauge modeling (Tay et al., 2020; Wu et al., 2021; Zhang et al., 2021, 2022; Saparov and He, 2022; Shi et al., 2023; van der Poel et al., 2023; Eldan and Li, 2023). Aside from being focused on the "smallest" and "purest" compositional unit of sequential reasoning (see Section 3.2), FFLM is distinguished by a few factors:

- **"$L_\infty$" objective:** Unlike usual benchmarks, we consider any model with less than $100\%$ accuracy as exhibiting a *reasoning error*. Aside from the motivation of completely eliminating hallucinations, we argue that this stringent notion of correctness is needed to avoid error amplification when flip-flops are embedded in more complex networks; see Appendix A.1 and Liu et al. (2023).

- **Parametric, procedurally generated, and generalizable:** Our empirical study precisely quantifies long-tail errors via a large number of replicates over the randomness of both model initialization and data generation. This methodology is easily adapted and rescaled (by adjusting $T, \mathbf{p}$, and other difficulty knobs) to probe language models of any size.

We provide an expanded discussion of related literature in Appendix A.2.

## 2    Background and notation

Modern language models are powered by *sequence-to-sequence (seq2seq) neural networks* $f_\theta :$ $\mathbb{R}^{T \times d} \to \mathbb{R}^{T \times d}$, which transduce sequences of vectors according to internal computations determined by the inputs as well as trainable parameters $\theta$. When equipped with mappings to and from symbolic tokens (an "embedding layer" $E : [M] \to \mathbb{R}^d$ (here, $M \in \mathbb{N}$ is the *vocabulary size*) and classification layer $W : \mathbb{R}^d \to \Delta([M])$, shared across positions), $W \circ f \circ E : [M]^T \to \Delta([M])^T$ can represent an autoregressive generative model of a joint distribution over tokens $x_{1:T} \in [M]^T$, where the output at the $t$-th position gives the estimated next-symbol probabilities $\widehat{\Pr}[x_{t+1} = \cdot \,|\, x_{1:t}]$. The overarching challenge of statistical language modeling is to fit complex distributions such as natural language; recurrent (Elman, 1990; Hochreiter and Schmidhuber, 1997; Wu et al., 2016) and self-attention-based (Vaswani et al., 2017) architectures have shown remarkable capabilities in fitting the seq2seq functions necessary for fluent linguistic parsing and reasoning.

**Recurrent inductive biases, attention, and length generalization.** To correctly process uniform (i.e. fixed-description-length) algorithmic computations on arbitrarily long sequences, it is natural to embed recurrences within a seq2seq network. Imitating the recurrent nature of the Turing machine, one can hope for RNNs to learn representations of the desired looped computations (Sutskever et al., 2013; Graves et al., 2014; Linzen et al., 2016). However, the key innovation in the Transformer architecture is a non-recurrent *self-attention* module.[4] Various works have noted that **Transformers and RNNs learn qualitatively different solutions**, discussing potential ways to reconcile these nuanced discrepancies (Dehghani et al., 2018; Abnar et al., 2020; Liu et al., 2023).

---

[4]We define self-attention in Appendix C. For in-depth breakdowns of the architectural components of Transformers, see (Vaswani et al., 2017; Phuong and Hutter, 2022; Edelman et al., 2022).

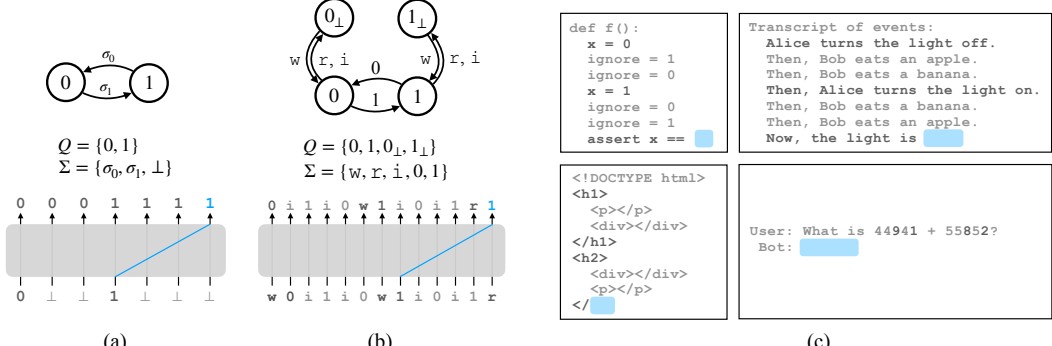

Figure 2: Elementary objects and examples associated with flip-flop languages. (a) the 2-state flip-flop machine (elided transitions are self-loops). (2) A 4-state automaton which processes flip-flop languages (implying the existence of a small RNN). (c) Simple examples of sequential prediction tasks which require processing a flip-flop language.

## 3 Flip-flop automata and the FFLM task

### 3.1 Definitions

For any even number $T \geq 4$, we define a flip-flop string as a sequence of symbols $\{\texttt{w}, \texttt{r}, \texttt{i}, 0, 1\}^T$, which have the semantics of *instructions* (`write`, `read`, `ignore`) and *data* (one bit). A valid flip-flop string consists of alternating pairs of instructions and data (e.g. "`w 0 i 1 i 0 r 0`"), for which every symbol following a `r` instruction must be equal to the symbol following the most recent `w`; thus, "`w 0 i 1 w 1 r 0`" is not a legal flip-flop string. These sequences can be viewed as correct execution transcripts of a program which can (perhaps occasionally) `write` to a single bit of memory, and always correctly `read`s its contents. All sequences are required to begin with `w` and end with `r`.

There are many possible choices of (probabilistic) *flip-flop languages*, which are distributions over valid flip-flop strings. We define a canonical family of them: let $\mathsf{FFL}(T, \mathbf{p})$ be the distribution over length-$T$ flip-flop strings, parameterized by $\mathbf{p} = (p_{\texttt{w}}, p_{\texttt{r}}, p_{\texttt{i}}) \in \Delta(\{\texttt{w}, \texttt{r}, \texttt{i}\})$, such that:

(i) The first instruction $x_1$ is always `w`, and the last instruction $x_{T-1}$ is always `r`.

(ii) The other instructions are drawn i.i.d. according to $(p_{\texttt{w}}, p_{\texttt{r}}, p_{\texttt{i}})$ with $p_{\texttt{i}} = 1 - p_{\texttt{w}} - p_{\texttt{r}}$.

(iii) The nondeterministic data symbols (paired with `w` or `i`) are drawn i.i.d. and uniformly.

We are interested in whether language models can learn a flip-flop language from samples, which we define as processing the `read` operations *perfectly*. Two variants of the autoregressive language modeling task can be defined on this distribution:

- **Generative ("noisy") mode:** Estimate the conditional next-token distribution $\Pr[x_{t+1}|x_{1:t}]$, for each $t = 1, \ldots, T - 1$. In this mode, the sequences can be treated as drop-in replacements for natural text in GPT-style training. Generative FFLMs can be evaluated by checking their completions on prefix "prompts" (e.g. "... `w 0 i 1 i 1 i [?]`").

- **Deterministic ("clean") mode:** Predict only the continuations which are deterministic: correctly output $x_{t+1}$ only for the prefixes $x_{1:t}$ such that $x_t = \texttt{r}$. At the cost of a slight departure from vanilla language modeling, this setting isolates the long-range memory task. It is similar to the non-autoregressive flip-flop monoid simulation problem discussed in Liu et al. (2023), with limited supervision. [5]

These tasks naturally embed the capability of simulating the *flip-flop*, a machine which memorizes a single bit (see Figure 2a,b for closely related variants).[6] It is easy to see that recurrent networks and

---

[5]We observe similar behaviors across these two settings (see Appendix B.2), but we report results on the "clean" setting in this paper. Predicting the non-deterministic tokens is irrelevant to the memory task at hand.

[6]A further discussion of the rationale for this specific manifestation of flip-flop sequence processing is deferred to Appendix A.3.

2-layer Transformers (see Proposition 2) *can* both represent FFLM parsers. The question of whether they *do*, especially from less-than-ideal data, turns out to be extremely subtle, and is the subject of the remainder of this paper.

## 3.2   Why focus on the flip-flop?

The most immediate rationale for this synthetic benchmark is that flip-flop simulation (maintaining memory in a sequence) is a direct necessity in many reasoning settings (see Figure 2c). It is a special (depth-1) case of Dyck language processing (Chomsky and Schützenberger, 1959; Yao et al., 2021; Zhao et al., 2023), which is necessary for parsing recursive grammars. It also captures certain structures in code or language tasks, such as tracking semantic changes (Miceli-Barone et al., 2023) or ignoring irrelevant contexts in general (Tafjord et al., 2020; Ho et al., 2020; Shi et al., 2023). The BIG-Bench suite of reasoning benchmarks (Srivastava et al., 2022) contains many tasks which require maintaining a discrete state over a sequence of transitions. Thus, more than a toy model, flip-flop languages are embedded verbatim within many sequence processing tasks. We offer some additional perspectives below.

**Algebraic properties and expressive power.**   Flip-flops are the computational building blocks of memory. The *flip-flop monoid* $\mathcal{F}$ (Definition 1), an algebraic encoding of a flip-flop's dynamics, is the smallest monoid whose operation is both *non-commutative* and *non-invertible*. $\mathcal{F}$ plays an essential role in the Krohn-Rhodes theory of automata and semigroups (Rhodes et al., 2010), whose central structure theorem (Krohn and Rhodes, 1965; Zeiger, 1967; Eilenberg, 1974) implies that a constant-depth cascade of parallel flip-flops simulates *all* group-free finite-state automata. Thus, in a rigorous sense, the robust learning of flip-flops is not only a *necessary* condition for reasoning, but a *sufficient* condition for a wide class of algorithmic capabilities.

**Intended functionality of attention.**   One can also appeal to the origin of attention mechanisms (Bahdanau et al., 2014; Luong et al., 2015; Vaswani et al., 2017): attention was specifically designed to *attend* to[7] (i.e. selectively retrieve and copy) data over long-range dependencies. Indeed, it is easy to verify that a single attention head can perform the required lookup (see Proposition 2). It is thus logical to ask how well a purely attention-based architecture performs this elementary operation.

## 4   Attention glitches: a long tail of errors for Transformer FFLMs

In our main set of synthetic experiments, we train neural language models to generate strings from the flip-flop language $\mathsf{FFL}(T = 512, \mathbf{p} = (0.1, 0.1, 0.8))$ (for short, $\mathsf{FFL}(p_\mathtt{i} = 0.8)$), [8] and probe whether the networks robustly learn the language. Although every valid flip-flop string is supported in this distribution, some sequences are far rarer than others; we measure tail behavior via probes of extrapolation, defined here as out-of-distribution evaluations which amplify the probabilities of the rare sequences. To create these "challenging" sequences, we sample $> 3 \times 10^5$ sequences from $\mathsf{FFL}(0.98)$ (containing unusually many "sparse" sequences with mostly `ignore` instructions), as well as $\mathsf{FFL}(0.1)$ (many "dense" sequences). Training and evaluating the `read` accuracies of Transformer models of various sizes, as well as a recurrent LSTM model, we find the following (see Figure 3):

(R1) **Transformers exhibit a long, irregular tail of errors.** Such errors occur on both sparse and dense sequences. Further, a model's out-of-distribution test error varies significantly between random seeds (initializations as well as stochastic minibatches), and even between iterates within the same training run.

(R2) **LSTMs extrapolate perfectly.** In stark contrast, with 20 times fewer training samples and iterations, a 1-layer recurrent model achieves $100\%$ accuracy, on 100 out of 100 runs.

**Data release.**   For reproducibility, we publish this synthetic data at `https://huggingface.co/datasets/synthseq/flipflop`: 16M $\mathsf{FFL}(0.8)$ training sequences, 16K $\mathsf{FFL}(0.8)$ in-distribution

---

[7]What this formally entails for representation and generalization is a topic of recent theoretical inquiry (Edelman et al., 2022; Wei et al., 2022a).

[8]In the remaining of the paper, we will use $\mathsf{FFL}(p_\mathtt{i})$ as a shorthand, with $T = 512$ and $p_w = p_r = \frac{1-p_i}{2}$.

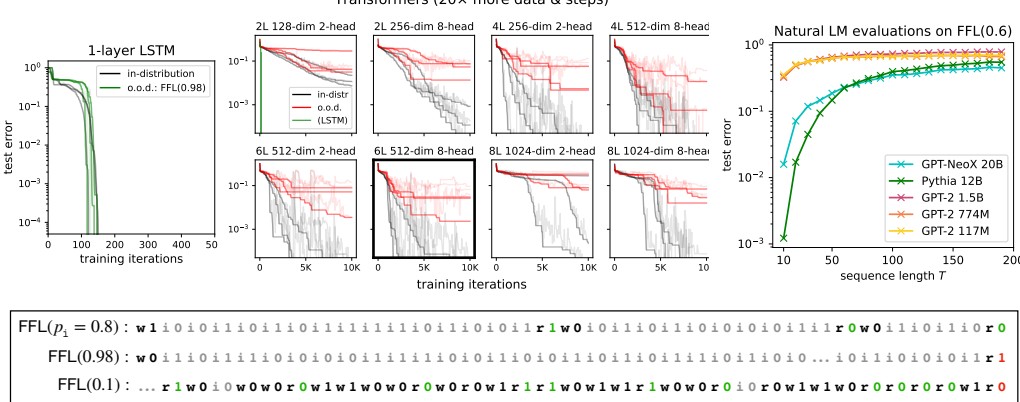

Figure 3: *Top:* Training curves of recurrent (left) vs. Transformer (center) architectures on FFLM, with *best-so-far* evaluation errors highlighted for clarity. **Transformers fail to extrapolate robustly** to the long tail of long-range dependencies, even on this extremely simple task of remembering one bit. The boxed box contains our chosen 6-layer 19M-parameter canonical baseline model. We find that the ability to complete flip-flop language prompts emerges in natural language models, but is not robust (right). *Bottom:* examples from the sparser FFL(0.98) and denser FFL(0.1) distributions, causing distinct (*long-range* and *short-range*) failure modes for the baseline Transformer model.

test sequences, 160K sparse o.o.d. sequences from FFL(0.98), and 4K FFL(0.1)) dense o.o.d. sequences from FFL(0.98). Our data pipelines can be replicated *exactly* by taking the appropriate prefix slices of these datasets.

As a counterpart to these findings, we observe similar anomalies in real LLMs, when prompted to complete natural textual embeddings (Figure 2, top right) of flip-flop tasks:

(R3) **10B-scale natural LMs can correctly process flip-flop languages, but not robustly.** Beyond a certain scale, natural language models can learn to process (natural embeddings of) flip-flop languages from in-context demonstrations. However, this emergent capability is not robust: there exist rare `read` errors, whose probabilities amplify as the sequence length $T$ grows. We provide details for the few-shot evaluation protocol in Appendix B.2.1.

### 4.1 Multiplicity of mechanisms for attention glitches

What failure mechanisms account for these reasoning errors, which occur for both short- and long-range dependencies? The model capability is not a concern as discussed earlier (see Proposition 2). In this section, we discuss how Transformer self-attention modules, when tasked with representing flip-flops, can exhibit multiple (perhaps mutually entangled) failure mechanisms. The accompanying propositions are proven in Appendices C.2 and C.3.

**An insufficient explanation: implicit $n$-gram models.** As a warmup, consider a language model $\widehat{\Pr}[x_{t+1}|x_{\leq t}]$ which only depends on the $n$ most recent tokens in the context. Then, if $n \ll \frac{1}{1-p}$, the bulk of $\widehat{\Pr}$'s predictions on FFL($p_{\mathtt{i}} = p$) can be no more accurate than random guessing. This recovers one qualitative trend (degradation of accuracy with dependency length) observed in the experiments. However, this cannot fully explain our findings: it fails to account for the incorrect predictions on dense sequences. Furthermore, the Transformers' outputs on FFL(0.98) are *mostly* correct; their accuracies on very long-range dependencies are nontrivial, despite not being perfect. There must therefore be subtler explanations for these errors.

**Lipschitz limitations of soft attention.** Moving to finer-grained failure mechanisms, a known (Hahn, 2020; Chiang and Cholak, 2022) drawback of soft attention is that its softmax operation can be "too soft"—for any weight matrices with fixed norms, the attention gets "diluted" across positions as the sequence length $T$ increases, and can fail to perform an intended "hard selection" operation. We provide a formal statement and proof (Proposition 3) in Appendix C.2.

**Difficulty of non-commutative tiebreaking.** Can we simply robustify soft attention by replacing it with hard attention? We present a brief analysis which suggests that even hard attention can be brittle. In a stylized setting (one-layer models with linear position encodings), we show that self-attention can *confidently attend to the wrong index*, unless the weight matrices precisely satisfy an orthogonality condition (Proposition 4). This suggests the existence of *spurious local optima*, which we do not attempt to prove end-to-end; however, we provide supporting empirical evidence in the experiments in Appendix C.3.

## 5 Mitigations for attention glitches

In this section, we investigate various approaches towards eliminating the long tail of reasoning errors exhibited by Transformer FFLMs. We select the 19M-parameter model (which has $L = 6$ layers, $d = 512$ embedding dimensions, and $H = 8$ heads) from Section 4 as a canonical baseline, and conduct precise evaluations of various direct and indirect interventions.

### 5.1 Effects of training data and scale

**Ideal solution: improving data coverage.** Prior work has made clear that data significantly impacts the performance (Schuhmann et al., 2022; Eldan and Li, 2023). Hence, we begin by examining what is perhaps the most obvious solution: removing the need for out-of-distribution extrapolation, by training directly on more diverse examples. Indeed, we verify that this works near-perfectly:

(R4) **Training on rare sequences works best, by a wide margin.** By training on a uniform mixture of FFL distributions with $p_\mathtt{i} = \{0.9, 0.98, 0.1\}$, the baseline architecture reliably converges to solutions with significantly fewer errors on each of these 3 distributions (teal violins in Figure 4). In 6 out of 25 runs, we did not detect a single error.

This should not be surprising, in light of the realizability of flip-flops by self-attention (and, more generally, the existence of shortcuts functionally identical to RNNs (Liu et al., 2023)), and corroborates similar conclusions from (Zhang et al., 2021). We also find that weaker improvements emerge by straightforwardly increasing scale parameters in the model and training pipelines:

(R5) **Resource scaling (in-distribution data, training steps, network size) helps.** However, the improvements are orders of magnitude smaller than those in (R4), and we observe tradeoffs between sparse- and dense-sequence extrapolation; see the blue violins in Figure 4.

Another class of direct solutions is to *externalize the chain of thought* (CoT): train (or finetune, or prompt) the model to explicitly output the intermediate reasoning steps (Nye et al., 2021; Wei et al., 2022b). We do not investigate this strategy in this paper, and note that prior work has provided sufficient evidence to affirm its efficacy in inducing the robust learning of recurrences on long synthetic sequences (Anil et al., 2022; Zhou et al., 2022; Liu et al., 2023). Even when applying CoT in practice, we believe attention glitches may still occur, as flip-flops operations may be embedded within a single indivisible reasoning step. Thus, the focus of this work is to isolate and mitigate this intrinsic architectural issue. We provide additional references and discussion in Appendix A.2.

### 5.2 Indirect algorithmic controls: a bag of regularization tricks

The interventions listed in Section 5.1 are all potentially practical, and may shed light on how closed-domain LLM hallucinations will diminish with data quality, scale, and improved inference strategies. However, it is not always *feasible* to implement these fixes under resource constraints (especially data). We next investigate an orthogonal design space, of how to robustify the *internal* memory mechanisms of neural sequence models. Note that the exceptionally strong extrapolative performance of the LSTM provides a "skyline", showing the possibility of far more robust architectures than the Transformer (in the flip-flop setting, with this restricted set of considerations).

**Standard regularization heuristics.** There is a large array of not-fully-understood algorithmic tricks for "smoothing" the behavior of LLMs. We test the extrapolative behavior of models trained with weight decay and dropout (at the attention, feedforward, and embedding layers), as well as a host of algorithmic choices known to modulate generalization (batch sizes, learning rates, optimizer

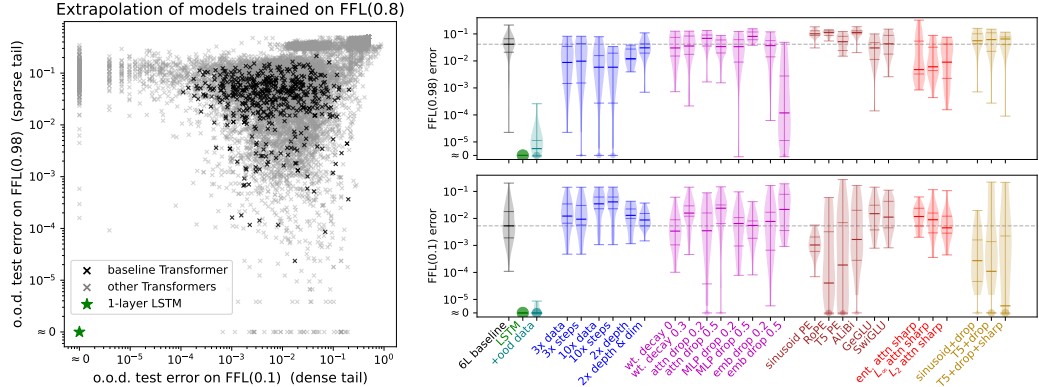

Figure 4: A long tail of flip-flop errors for 10,625 Transformer models. *Left:* Out-of-distribution evaluations for all models; some algorithmic choices help substantially (note the logarithmic axes), but **nothing we tried, aside from training on o.o.d. data, could fully eliminate attention glitches**. *Right:* Effects of individual architectural and algorithmic choices on both types of extrapolation (sparse and dense sequences). Some configurations reduce attention glitch rates by orders of magnitude. Horizontal marks denote {min, 25%, median, 75%, max} test errors on $> 3 \times 10^5$ predictions, over 25 replicates (500 for the baseline model). Dots at the bottom indicate runs with 0 error.

hyperparameters, position embeddings, activation functions). Due to the extreme variability noted in (R1), we quantify effects on extrapolation by training and evaluating at least 25 replicates for each choice under consideration.

**Attention sharpening: a non-standard regularization technique.** Inspired by the "diluted hard attention" calculation in Section 4.1, and the fact that the attention heads of trained models do not attend sharply (see Figure 5), we train Transformer models with *attention-sharpening regularizers*:[9] during training, for attention weights $\alpha \in \Delta([T])$, adding differentiable loss terms which encourage sparsity (e.g. the mixture's entropy $H(\alpha)$, or negative $p$-norms $-\|\alpha\|_2, -\|\alpha\|_\infty$).

(R6) **Many algorithmic choices influence extrapolative behaviors.** We find that some architectural variants and regularization tricks have orders-of-magnitude effects on the out-of-distribution performance of Transformers; see the purple, brown, red, and gold violins in Figure 4 (right). Our strongest improvements on sparse sequences are obtained by large (0.5) embedding dropout and attention-sharpening losses; on dense sequences, non-trainable position embeddings are the most helpful.

(R7) **Despite many partial mitigations, nothing eliminates attention glitches entirely.** The scatter plot in Figure 4 (left) gives an overview of our entire search over architectures and hyperparameters, showing (dense-sequence error, sparse-sequence error) pairs for *every* model we trained. We found it extremely difficult to find a setting that reliably produces Transformer models with simultaneous improvements over the baseline on sparse and dense sequences. Recall that it is trivial to do so with an LSTM model.

## 5.3 Preliminary mechanistic study of the trained networks

In this section, we move to a simpler setting to gain finer-grained understanding of how sparsity regularization affects the learned solutions. Specifically, we look at the task of *simulating the flip-flip automaton* (Definition 1), whose inputs consist of $\{\sigma_0, \sigma_1, \bot\}$ as two types of `write` and 1 no-op. This task (elaborated in Appendix A.1) can be solved by a 1-layer Transformer with a single attention head which attends sparsely on the most recent `write` position. It also serves as a building block for more complex tasks (Liu et al., 2023), hence observations from this simple setup can potentially be useful in broader contexts.

---

[9]While less popular, such losses have been used to sparsify dependencies in similar contexts (Zhang et al., 2018; Sukhbaatar et al., 2021).

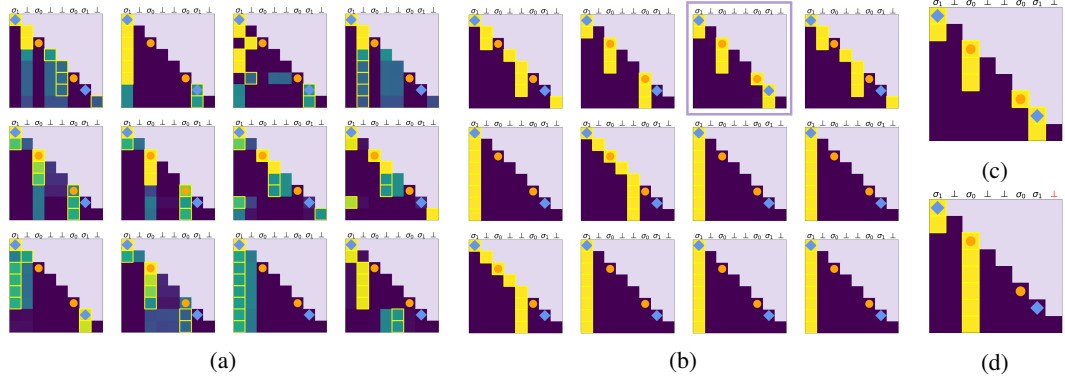

Figure 5: Causal attention patterns for flip-flop simulation (Definition 1); orange dots / blue diamonds mark the positions of write tokens $\sigma_0$ / $\sigma_1$. (a),(b) are subselected respectively from a regular (non-sparse) and a sparse multi-layer model (details in Appendix B.5). (c), (d) are from two 1-layer 1-head models. The attention pattern highlighted by the purple box in (b) coincides with the "ideal" attention pattern in (c). However, sparse models can be wrong, as shown in (d) (error marked in red).

Figure 5 shows examples of attention patterns on the flip-flop simulation task, subselected from 6-layer 8-head models trained with and without attention-sharpening regularization. It is evident that the attention patterns of the sparse model are less complex and easier to interpret compared to those of the un-regularized model. For example, we can identify one head in the sparse model that exactly coincide with the attention pattern[10] that an "ideal" 1-layer 1-head model implements (Figure 5c).

(R8) **Attention-sharpening regularizers successfully promote hard attention, but errors persist.**
As mentioned in (R7), attention-sharpening regularization cannot fully eliminate the sporadic errors, which are partially induced by the complexity and redundancy of attention patterns. Moreover, sharpened attention can induce additional failure modes, such as confidently attending to incorrect `write` positions. An example is demonstrated in Figure 5d, where the attention focuses on an initial `write`, likely caused by the fact that earlier positions are overemphasized due to the use of causal attention masks. Another example occurs in length generalization, where the attention is correct at positions earlier in the sequence, but starts to confidently focus on wrong positions as it moves towards later positions (Proposition 4).

In a similar spirit to concurrent work on generating Dyck languages (Wen et al., 2023) (a more complex capability which also requires parallel simulation of memory registers), these glitchy solutions point to a concrete obstruction to mechanistic interpretability. Due to factors such as overparameterization, spurious solutions, and the opaqueness of optimization dynamics, **learned neural implementations of algorithms generally do not coincide with "ideal", "minimal", or "natural" theoretical constructions**. Details for these experiments and further discussion are provided in Appendix B.5.

## 6 Conclusion and future challenges

We have introduced *flip-flop language modeling* (FFLM), a synthetic benchmark for probing the fine-grained extrapolative behavior of neural sequence models, based on a one-bit memory operation which forms a fundamental building block of algorithmic reasoning. Despite being able to realize this operation trivially, **Transformer models do not extrapolate robustly**: they exhibit a long tail of sporadic reasoning errors, which we call *attention glitches*. Through extensive controlled experiments, we find that many algorithmic mitigations can reduce the frequency of attention glitches, but **only recurrence and training on longer-tailed data work perfectly**. FFLM provides a concrete and minimalistic setting in which Transformers are far inferior to recurrent sequence models, with respect to multiple criteria (efficiency, stability, and extrapolation).

---

[10]While it is well-known that attention patterns can be misleading at times (Jain and Wallace, 2019; Bolukbasi et al., 2021; Meister et al., 2021), they do provide upper bounds on the magnitude of the dependency among tokens. These upper bounds are particularly useful in the case of (1-)sparse attention: a (near) zero attention weight signifies the absence of dependency, which greatly reduces the set of possible solutions implemented.

**What does this entail about hallucinations in natural LLMs?** The motivating issue for this work is the phenomenon of "closed-domain hallucinations" in non-synthetic LLMs (e.g. the errors demonstrated in Figure 1). We hypothesize that attention glitches occur in the internal algorithmic representations of Transformer models of natural language, and that they account for (a non-negligible portion of) the reasoning errors encountered in practice. To our knowledge, this is the first attempt to attribute model hallucinations to a systematic architectural flaw in the Transformer. However, confirming or refuting this hypothesis is far outside the scope of this paper; the opaque indirections and lack of adequate controls on the training data present significant methodological challenges. Even precisely articulating this hypothesis leaves degrees of freedom which are difficult to resolve; see the discussion in Appendix A.4. We therefore leave these topics for future work.

**Paths to hallucination-free Transformers?** Our findings suggest that in the near term, there are many mutually-compatible approaches for reducing the frequency of attention glitches: data (particularly with high diversity), scale, and various forms of regularization. Yet, the strikingly outsized benefit of replacing the Transformer with an LSTM network suggests that *architectural* innovations towards the same ends are well worth examining. Obtaining a practical best-of-both-worlds architecture is a grand open challenge, for which new recurrent designs (Katharopoulos et al., 2020; Dao et al., 2022; Peng et al., 2023; Anonymous, 2023) show great promise. Note that we do not make the claim that recurrent architectures are the only ones which can extrapolate robustly.[11]

**Broader impacts and limitations.** This work is inherently foundational, and focuses on precise measurements of generalization in an idealized setting; see Appendix A.4 for a discussion of the limitations this entails. By introducing methodologies to isolate, measure, and control the long tail of reasoning errors in neural sequence models, we hope that this work will contribute to the systematic and principled discovery of LLM pipelines with improved factual reliability. Such improvements may result in unintended downstream consequences, such as higher-fluency malicious content generation.

## Acknowledgements

We would like to thank Yuchen Li for helpful discussions and comments on the paper.

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

# Appendix

## Table of Contents

# A Deferred background and discussion

## A.1 Flip-flop terminology and history

The *flip-flop automaton*[12] is a two-state machine which remembers a single bit of memory, and enables retrieval of this bit. More precisely, the flip-flop automaton (illustrated in Figure 2(a)) is defined as:

**Definition 1** (Flip-flop automaton). *A flip-flop automaton* $\mathcal{A} = \{Q, \Sigma, \delta\}$ *is defined with state space* $Q = \{0, 1\}$*, input alphabet* $\Sigma = \{\sigma_0, \sigma_1, \perp\}$*, and transition function* $\delta : Q \times \Sigma \to Q$ *where*

$$\begin{cases} \delta(q, \sigma_0) & = 0, \\ \delta(q, \sigma_1) & = 1, \qquad \forall q \in \{0, 1\}. \\ \delta(q, \perp) & = q; \end{cases}$$

The semantics of the input symbols can be intuitively be identified with "write 0", "write 1", and "do nothing". This mathematical object is named after a type of electronic circuit which can store a single bit of state information (Eccles and Jordan, 1918, 1919); such physical constructions appear ubiquitously in electrical engineering as the building blocks of memory. The task of interest in Section 5.3 is *simulating the flip-flop automaton*: the model takes as input a sequence of $\sigma_1, \sigma_2, \cdots, \sigma_T \in \Sigma$, and learns to output the corresponding states $q_t \in Q$ for each $t \in [T]$ after processing inputs $\sigma_{1:t}$.

Naturally associated with the flip-flop automaton is its *transformation monoid*, the closure[13] of its *state transformations* $\delta(\,\cdot\,, \sigma) : Q \to Q$ under function composition. Identifying each symbol with its state transformation map, we can compute the multiplication table of this monoid ($f \circ g$ for every pair of transformations $f, g$):

|                | $g = \sigma_0$ | $g = \sigma_1$ | $g = \perp$ |
| -------------- | -------------- | -------------- | ----------- |
| $f = \sigma_0$ | $\sigma_0$     | $\sigma_0$     | $\sigma_0$  |
| $f = \sigma_1$ | $\sigma_1$     | $\sigma_1$     | $\sigma_1$  |
| $f = \perp$    | $\sigma_0$     | $\sigma_1$     | $\perp$     |

This algebraic object is called the *flip-flop monoid* $\mathcal{F}$. Its binary operation $\circ$ is clearly (1) non-invertible (intuitively: the history of the bit cannot be recovered after a "memory write") and (2) non-commutative (the order of "write" operations matters); it also has (3) an identity element $\perp$ (which does nothing to the memory bit). By enumeration of smaller objects, it can be seen that $\mathcal{F}$ is the smallest monoid (in terms of order $|\mathcal{F}|$, or fewest number of automaton states $|Q|$) which has properties (1)-(3).

The flip-flop monoid plays a special role in the algebraic theory of automata (Rhodes et al., 2010): flip-flops can be cascaded to represent more complex functions. In particular, the celebrated Krohn-Rhodes theorem (Krohn and Rhodes, 1965) gives a "prime decomposition" theorem for *all* finite semigroups (associative binary operations), representing them as alternating wreath products of flip-flop monoids and finite simple groups. Further developments (Zeiger, 1967; Eilenberg, 1974) have interpreted this theorem as a structural reparameterization of any finite-state automaton into a feedforward hierarchy of simple "atomic" machines (namely, flip-flops and permutation semiautomata). Basic quantitative questions (e.g. "which functions of $n$ variables can $L$ layers of $\text{poly}(n)$ flip-flops represent?") have proven to be extremely hard to resolve; these are studied by the theories of Krohn-Rhodes complexity and circuit complexity.

It was noted by Barrington and Thérien (1988) that these reparameterizations of finite-state automata entail the existence of parallel algorithms (i.e. shallow, polynomial-sized circuits) for sequentially executing finite-state recurrences (thus, processing formal languages) on sequences of length $T$. More recently, Liu et al. (2023) establish implications for shallow Transformer neural networks: they show that they can size-efficiently (with depth $O(\log T)$ and parameter count $\Theta(T)$; sometimes both

---

[12]Sometimes, a distinction is made between a semiautomaton $(Q, \Sigma, \delta)$ and an automaton, which is a semiautomaton equipped with a (not necessarily invertible) mapping from states to output symbols. We do not make such a distinction; we equip a semiautomaton with the output function which simply emits the state $q$, and use "automaton" to refer to this dynamical system.

[13]In this case, the closure is the same as the generator set: no functions distinct from $\sigma_0, \sigma_1, \perp$ can be obtained by composing these three functions. This is not true for a general automaton.

improvable to $O(1)$) realize these parallel algorithms, and that standard gradient-based training can empirically learn $\ll T$-layer solutions on a variety of "hard" automata (e.g. composing a sequence of $T$ 5-element permutations; multiplying $T$ unit quaternions). Here, the role of the flip-flop monoid is essential: it provides a natural way to think about the role of a single self-attention head in a hierarchy of indirections, in order to learn a depth-constrained parallel implementation of a sequential algorithm.

## A.2 Additional related work

**Relevant challenges in NLP: hallucinations and long-range dependencies.** The empirical literature is rife with corroborations that neural language models have trouble with robustly fitting long-range memory and multi-step reasoning (Khandelwal et al., 2018; Sun et al., 2021; Sukhbaatar et al., 2021; Malkin et al., 2022; Saparov and He, 2022; Orvieto et al., 2023; Creswell et al., 2023). Such failures can result in "hallucinations": incorrect outputs which either directly contradict factual input in the context, or contain information absent in the context (Ji et al., 2023).

Hallucination can be attributed to various factors, such as the noisiness in data sources (Dhingra et al., 2019; Dziri et al., 2022), imperfect encoding/decoding (Parikh et al., 2020; Tian et al., 2019), or the discrepancy in training and evaluation setups (He et al., 2019). In particular, the most related to our paper are the characteristics inherent to the model itself. For example, prior work has found that Transformers tend to be biased towards information covered during training (Petroni et al., 2019; Longpre et al., 2021), a potential cause to their poor out-of-distribution performance, and may over-commit to their earlier errors (Zhang et al., 2023)

In terms of mitigation, various "external" methods (i.e. ones which do not modify the internal representations of the neural network) have been proposed to address some of the above factors, or post-processing model generations (Dziri et al., 2021; Chen et al., 2021), possibly utilizing several forward passes (Wang et al., 2022; Zheng et al., 2023). Another line of work that have gained much popularity and success is to incorporate explicit memory mechanisms, which we discuss next.

**Explicit memory mechanisms in Transformers.** Prior work has shown that augmenting the neural network with memory modules or knowledge base helps improve the performance on long-range texts (Khandelwal et al., 2019; Wu et al., 2022; Bertsch et al., 2023). An approach particularly effective for large-scale Transformers is to ask the model to output immediate reasoning steps to a "scratchpad" which the model subsequently processes (Nye et al., 2021; Wei et al., 2022b; Zhou et al., 2022; Anil et al., 2022; Shao et al., 2023), similar to writing to and reading from a memory tape. A particular way to interact with the scratchpad is to interlace every other token with an annotation of "as a reminder, this is the state" (Liu et al., 2023; Lanchantin et al., 2023), so that there are no more explicit long-range dependencies. However, this strategy is the same as the recurrent solution implementable by RNNs, and it does not always exist, especially when attention glitches occur in an internal component of the model.

**Transformers and algorithmic tasks.** Compared to real-world language datasets, synthetic tasks provide a cleaner and more controlled setup for probing the abilities and limitations of Transformers. Specific to algorithmic reasoning, Liu et al. (2023) puts a unifying perspective on the ability of small Transformers to succeed at tasks corresponding to algorithmic primitives. Specific tasks of interest include modular prefix sums (Hahn, 2020; Anil et al., 2022), adders (Nogueira et al., 2021; Nanda and Lieberum, 2022; Lee et al., 2023), regular languages (Bhattamishra et al., 2020; van der Poel et al., 2023), hierarchical languages (Yao et al., 2021; Zhao et al., 2023), and following a chain of entailment Zhang et al. (2022).

**Comparison with *Transformers Learn Shortcuts to Automata*.** Liu et al. (2023) study the parallel circuits efficiently realizable by low-depth Transformers. The authors identify *shortcut solutions*, which exactly replicate length-$T$ recurrent computations ("chains of algorithmic reasoning") in the absence of recurrence, with very few ($O(\log T)$; sometimes $O(1)$) layers. Their results contain a general structure theorem of *representability*, and preliminary positive empirics for *generalization* and *optimization*, demonstrating that Transformers can learn these shallow solutions via gradient-based training on samples. In contrast, the present work is a fine-grained study of the issue of generalization. Our main empirical contributions are a minimally sufficient setup (FFLM) and a

set of large-scale[14] controlled experiments, towards providing reasonable scientific foundations for addressing the unpredictable reasoning errors of LLMs.

### A.3  Why *this* flip-flop language?

Liu et al. (2023) (as well as our mechanistic interpretability experiments) use a purer instantiation of flip-flop sequence processing, in which the sequence-to-sequence network is tasked with *non-autoregressive transduction*: given the sequence of input symbols $\sigma_1, \ldots, \sigma_T$, output the sequence of states $q_1, \ldots, q_T$. This is most natural when studying the Transformer architecture's algebraic representations in their most isolated form.

Our autoregressive sequence modeling setting is a slight departure from this setting; we discuss its properties and rationale below.

- The autoregressive setting "type-checks" exactly with standard state-of-the-art autoregressive (a.k.a. causal, forward, or next-token-prediction) language modeling. This makes it more convenient and intuitive as a plug-and-play benchmark.

- The cost is a layer of indirection: the model needs to associate "instruction" tokens with their adjacent "data" tokens. This is a natural challenge for representation learning, and is certainly a necessary cursor for robust extrapolation on natural sequences that embed similar tasks (like those considered in Figure 2c). It is straightforward to remove this challenge: simply tokenize at a coarser granularity (i.e. treat (instruction, data) pair as a distinct vocabulary item).

- The multi-symbol (and variable-length-symbol, etc.) generalizations of the binary flip-flop language are more parsimonious. If there are $n$ instead of 2 tokens, this language can be encoded with $n + 3$ commands. Without the decoupling of "instruction" tokens from "data", the vocabulary size would scale suboptimally with $n$. In Figure 12, we provide a quick demonstration that attention glitches persist in the presence of larger vocabularies.

- The conclusions do not change: in smaller-scale experiments, we observe the same extrapolation failures between the autoregressive and non-autoregressive task formulations.

### A.4  Attention glitches in natural LLMs

In this section, we expand on the brief discussion from Section 6. At a high level, *we hypothesize that attention glitches cause (some) closed-domain hallucinations in Transformer models of more complex languages.* However, due to the fact that neural networks' internal representations evade simplistic mechanistic characterization, it is a significant challenge to formulate a rigorous, testable version of this hypothesis. We discuss the subtleties below.

First, we discuss a more general notion of attention glitches, of which the flip-flop errors considered in this papers are a special case. We define attention glitches as *failures of trained attention-based networks to implement a hard retrieval functionality perfectly.* To formalize this notion, there are several inherent ambiguities—namely, the notions of "hard retrieval" and "perfectly", as well as the granularity of "subnetwork" at which an attention glitch can be defined non-vacuously. The FFLM reasoning errors considered in this work provide a minimal and concrete resolution of these ambiguities. We discuss each of these points below:

- **Hard retrieval:** To succeed at FFLM, a network's internal representations must correctly implement the functionality of retrieving a single bit (from a sequence of bits, encoded unambiguously by the network), selected via the criterion of "most recent `write` position". This can be expanded into a richer functional formulation of hard attention, by generalizing the set of possible *retrieved contents* (a discrete set of larger cardinality, or, even more generally, a continuous set), as well as more complex *selection criteria* (e.g. "least recent position").

- **Ground truth:** Of course, to define "errors" or "hallucinations" in reasoning, there must be a well-defined *ideal* functionality. For FFLM, the notion of "closed-domain" reasoning and hallucinations is evident: the ideal behavior is for a model's outputs to coincide with that of the flip-flop machine on all input sequences. This straightforwardly generalizes to all formal languages, where the model is expected to correctly produce the deterministic outputs of automata

---

[14] $\sim 10^4$ 19M-parameter Transformers were trained in the making of this paper; see Appendix B.6.

| Input | GPT-3.5 | GPT-4 | Answer |
|---|---|---|---|
| 8493
+ 2357 | 10850 ✓ | 10850 ✓ | 10850 |
| 84935834
+ 23572898 | 108008732 ✗ | 108508732 ✓ | 108508732 |
| 9991999919909993
+ 6109199190990097 | 16111199100810090 ✗ | 16101199100890090 ✗ | 16101199110900090 |

Table 1: Examples (in Figure 1) of GPT variants on addition: While models tend to succeed at additions with a small number of digits, they (nondeterministically) fail at longer additions.

which parse these languages. By considering expanded notions of "ground truth", it is possible to capture other notions of model hallucinations (such as incorrectly memorized facts). Our work does not address open-domain hallucinations (i.e. where the model output contradicts real-world knowledge (OpenAI, 2023)), which may be unrelated to attention glitches.

- **Submodules:** Towards attributing implementations and errors to localized components of a network, it is impossible to provide a single all-encompassing notion of "localized component". This is a perennial challenge faced in the mechanistic interpretability literature. Our work considers two extremes: the entire network (in the main experiments, where we probe end-to-end behavior), and a single self-attention head (in Sections 4.1, 5.3 and Appendix B.5, in which we probe whether a single attention head can learn multiplication in the flip-flop monoid). Even when considering the same functionality, attention glitches can be considered for different choices of "submodule".[15] Our results reveal a key subtlety: in the presence of overparameterization (more layers and parallel heads than necessary according to the theoretical constructions), Transformers learn to process flip-flop languages via soft attention.

We expect that to effectively debug the full scope of LLM hallucinations, all of the above choices will need to be revisited, perhaps in tandem.

We hypothesize that the algorithmic reasoning capabilities of real LLMs (i.e. their ability to recognize, parse, and transduce formal symbolic languages) are implemented by *internal* subnetworks whose functionalities can be identified with generalizations of the flip-flop machine. To the extent that such modules exist, attention glitches (the failure of these modules to represent the flip-flop operations perfectly, due to insufficient training data coverage) cause sporadic end-to-end errors ("closed-domain hallucinations"). In this work, we have treated the case of *external* attention (where the task is to learn the flip-flop directly).

# B Full experimental results

## B.1 Details for LLM addition prompts (Figure 1)

These addition problem queries serve as a quick demonstration of (1) non-trivial algorithmic generalization capabilities of Transformer-based LLMs; (2) the brittleness of such capabilities: we directly address this type of reasoning error in this work. Table 1,2 show these queries and results in detail.

We emphasize that these examples were selected in an adversarial, ad-hoc manner; we do not attempt to formalize or investigate any claim that the errors made by larger models are at longer sequence lengths. We also cannot rule out the possibility that some choice of prompt elicits robust algorithmic reasoning (e.g. the prompting strategies explored in (Zhou et al., 2022)). The only rigorous conclusion to draw from Figure 1 is that of non-robustness: even LLMs exhibiting state-of-the-art reasoning continue to make these elementary errors for some unambiguous queries with deterministic answers. It was last verified on May 8, 2023 that GPT-4 (in its ChatGPT Plus manifestation) demonstrates the claimed failure mode.

---

[15]Beyond the two extremes considered in this work, some examples include "a subset of attention heads", "a subset of layers", and "a subspace of the entire network's embedding space".

| Input | GPT-3.5 | GPT-4 | Answer |
|---|---|---|---|
| 4491
+ 8759 | 13250 ✓ | 13250 ✓ | 13250 |
| 80087394
+ 63457948 | 143045342 ✗ | 143545342 ✓ | 143545342 |
| 5101611078665398
+ 8969499832688802 | 1.4071110911354202e+16 ✗ | 14071110911354196 ✗ | 14071110911354200 |

Table 2: More examples of GPT variants on addition: While models tend to succeed at additions with a small number of digits, they (nondeterministically) fail at longer additions.

## B.2 Extrapolation failures of standard Transformers (Section 4)

This section provides full details for our empirical findings (R1) through (R3).

**Architecture size sweep.** We consider a sweep over Transformer architecture dimensionalities, varying the three main size parameters. We emphasize that these are somewhat larger than "toy" models: the parameters go up to ranges encountered in natural sequence modeling (though, of course, far short of state-of-the-art LLMs).

- The *number of layers* (depth) $L \in \{2, 4, 6, 8\}$.
- The *embedding dimension* $d \in \{128, 256, 512, 1024\}$.
- The *number of parallel attention heads* per layer $H \in \{2, 4, 8, 16\}$. In accordance with standard scaling rules-of-thumb, each head's dimension is selected to be $d/H$.

**Other hyperparameter choices.** We use a sequence length of $T = 512$, again to reflect a typical length of dependencies considered by nontrivial Transformer models. We use a canonical set of training hyperparameters for this sweep: the AdamW (Loshchilov and Hutter, 2017) optimizer, with $(\beta_1, \beta_2) = (0.9, 0.999)$, learning rate $3 \times 10^{-4}$, weight decay $0.1$, 50 steps of linear learning rate warmup, and linear learning rate decay (setting the would-be 10001th step to 0). We train for 10000 steps on freshly sampled data, and choose a minibatch size of 16; consequently, the models in this setup train on 81,920,000 tokens.

**Training and evaluation data.** Unless otherwise stated, we train our models on online samples (fresh i.i.d. batches) containing moderate-length dependencies ($T = 512, p_{\mathtt{i}} = 0.8, p_{\mathtt{w}} = p_{\mathtt{r}} = 0.1$, or FFL(0.8) for short). We evaluate on the following held-out test sets, which are held the same across all training runs:

(i) *In-distribution:* $10^3$ sequences from the same distribution FFL(0.8), containing 26508 `read` instructions.
(ii) *Sparse tail:* $10^5$ sequences from FFL(0.98), containing 353875 `read` instructions.
(iii) *Dense tail:* 3000 sequences from FFL(0.1), containing 345781 `read` instructions.

When showing training curves, we evaluate errors only on (i) and the first $1\%$ of (ii) every 100 steps; the purpose of these is only to give a qualitative sense of instability, and affirm the presence of errors. For the main suite of experiments, we focus on reporting the high-precision glitch rate measurements on (ii) and (iii). Final errors on (i) are exactly 0 except for a small number of non-converged runs (2-layer architectures and badly tuned hyperparameters), so we omit these evaluations in favor of the more informative measurements which focus on the sparse and dense tails.

(R1) **Transformers exhibit a long, irregular tail of errors.** Figure 6 shows training curves for 3 replicates (random seeds) in each setting, while the scatter plot in the main paper shows variability of out-of-distribution accuracy across random seeds for the baseline setup. We find that Transformers sometimes succeed at extrapolation, but erratically.

(R2) **1-layer LSTM extrapolates perfectly.** We train a 1-layer LSTM (Hochreiter and Schmidhuber, 1997) network, with hidden state dimension 128 (for a total of 133K parameters), for 500

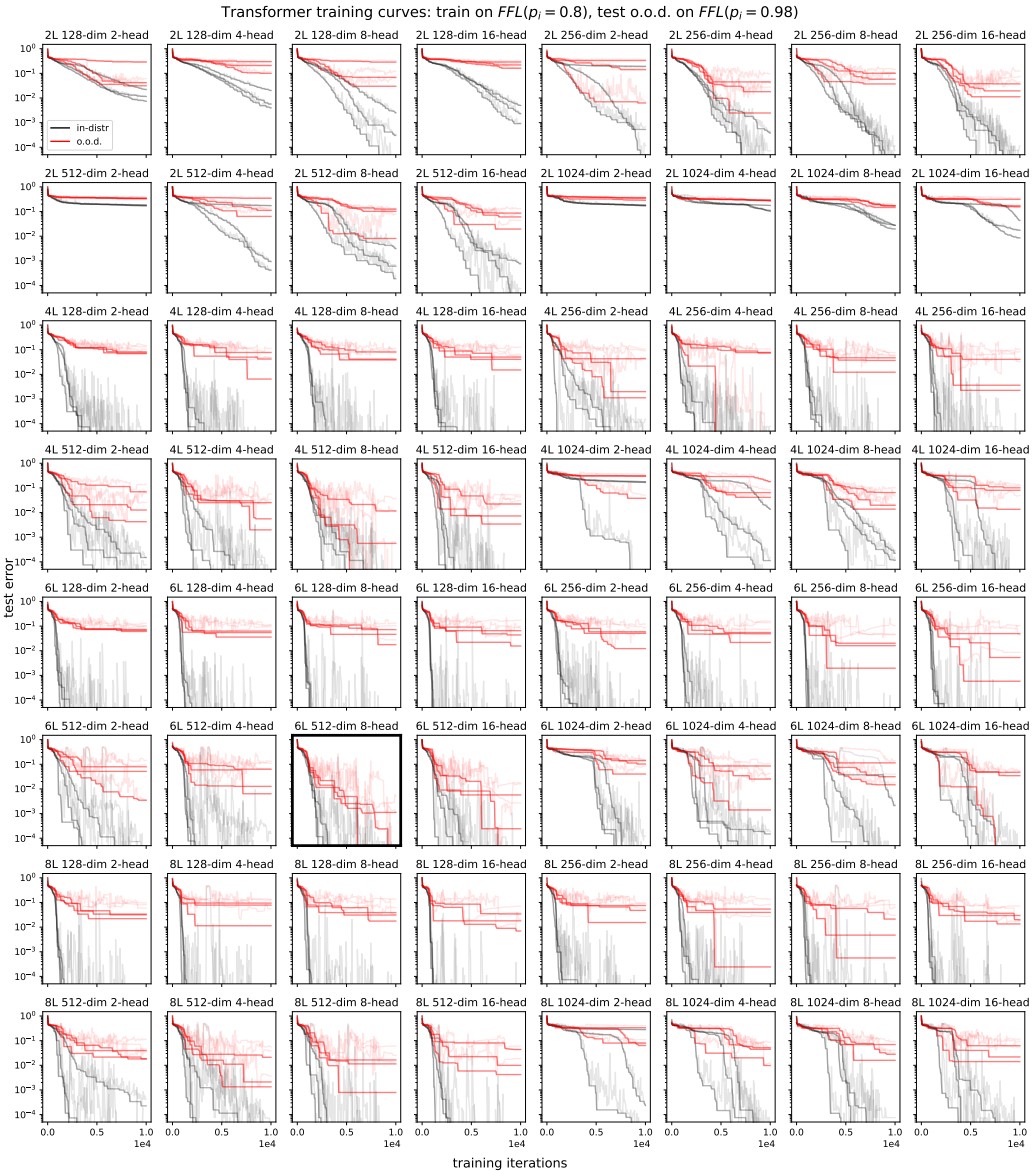

Figure 6: Examples of training curves over various Transformer architectures, ranging from 46K to 101M trainable parameters. We exhibit 3 (randomly selected) random seeds for each architecture. Lighter curves show raw error percentages, while solid curves denote the lowest error so far in each run. Notice the following: (1) **non-convergence of shallow models** (despite representability) (2) **failure of most runs to extrapolate** (i.e. reach 0% out-of-distribution error); (3) **high variability** between runs; (4) erratic, **non-monotonic progress** on out-of-distribution data, even when the in-distribution training curves appear flat; (5) **a small LSTM outperforms all of these Transformers** (see Figure 3). The bolded box represents our 19M-parameter baseline model.

steps with the same optimizer hyperparameters as above. The LSTM model achieves exactly $0$ final-iterate o.o.d. error, over $100$ out of $100$ replicates. In smaller experiments, we found that larger LSTMs (width up to $8192$, depth up to $4$) also extrapolate perfectly.

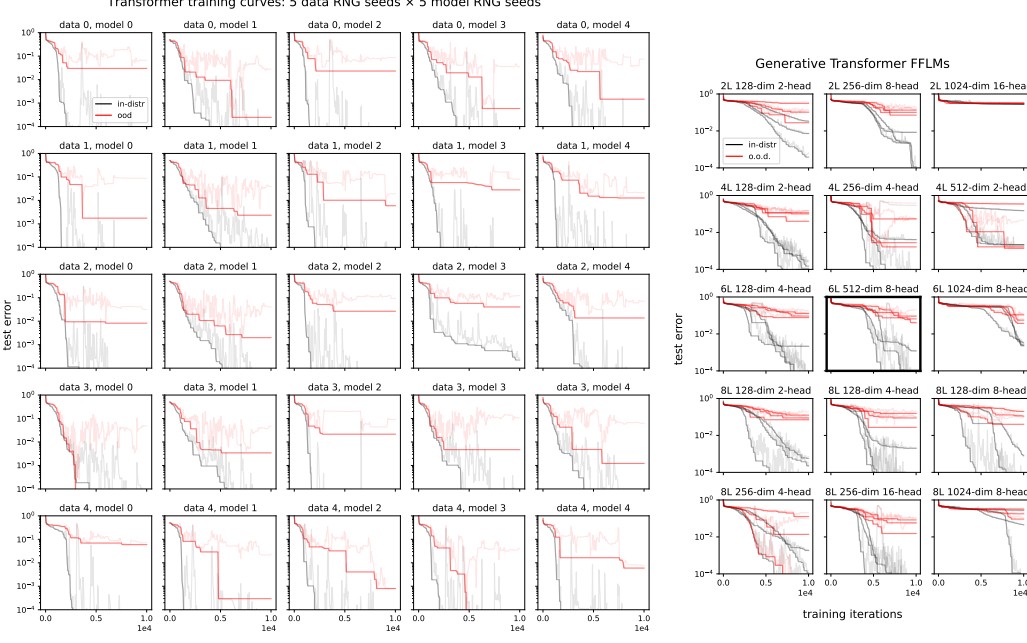

Figure 7: Additional training curves. *Left:* Identical baseline architecture, varying the 5 data seeds and 5 model seeds: models in the same row encounter the same sequence of data, while models in the same column start from identical initializations. **Both sources of randomness affect training dynamics and extrapolation**, and it is not clear which is more important. *Right:* Similar findings for models trained in "fully generative" mode (scoring on all tokens); baseline architecture is in the ⬜bolded box⬜.

**Canonical baseline.** We select the **6-layer, 512-dimensional, 8-head** architecture (with 19M trainable parameters) as our *canonical baseline* model: it is large in relevant dimensions[16] to real Transformers, while being small enough to allow for thousands of training runs at a reasonable cost. To fully understand the variability of this single architectural and algorithmic setup, we train and evaluate 500 replicates in this setting.

**Random data vs. random initialization.** Recent synthetic probes on the surprising behavior of deep neural nets on hard synthetic tasks (Barak et al., 2022; Garg et al., 2022) obtain additional insights by disentangling the effects of *data randomness* (i.e. the precise sequence of minibatches) vs. *model randomness* (e.g. random initialization and dropout). We provide a quick demonstration in Figure 7 (left) that *both sources of stochasticity matter*. We do not perform a more detailed investigation of their precise influence and roles.

**Fully generative setting: similar negative results.** As mentioned in Section 3.1, to capture a setting closer to standard autoregressive (sometimes called GPT-style) language modeling, we find a similar failure to extrapolate when models are trained to predict all tokens, rather than only the deterministic ones ($x_{t+1}$ such that $x_t = \mathtt{r}$). Figure 7 (right) exhibits some training curves for this setting, showing non-extrapolation, variability, and instability. We observe that training (to in-distribution convergence) takes slightly longer in this setting, and usually succeeds with the baseline architecture. We do not perform further controlled experiments in this setting.

### B.2.1 Evaluating real LLMs on flip-flops

We provide a quick corroboration that while LLMs in practice can perform in-context reasoning when the sequences are unambiguously isomorphic to a flip-flop language. We use the natural language

---

[16]Except the vocabulary size. In preliminary experiments, we obtained similar findings in the case of token spaces larger than $\{0, 1\}$.

**User**: Hi, let's play a game shall we! There are 3 instructions: write, read, and ignore. I will use w, r, i to denote each of them, respectively. Each instruction will be followed by a value (0 or 1), and the game is that: given a sequence of w, r, i and values (the sequence will always start with a w and end with a r), output the value get paired with the last w. Is the game clear?

**ChatGPT**: Yes, I understand the game. To clarify, here's what I understand:
1. "w" stands for "write" and will always be followed by a value (0 or 1).
2. "r" stands for "read".
3. "i" stands for "ignore" and will also be followed by a value (0 or 1).
The sequence always starts with a "w" and ends with a "r". My task is to output the value that is paired with the last "w" in the sequence before the final "r".
With that understanding, let's play! Please provide a sequence.

(a) Explaining FFLM to ChatGPT.

**User**: Here is a sequence: w 0 i 0 i 1 i 0 i 1 i 0 i 1 i 0 i 0 i 0 i 1 i 0 i 1 i 1 i 0 i 1 i 1 r.

**ChatGPT**: the "w" (write) operation only appears at the beginning, and the paired value is 0. So, despite many "i" (ignore) operations, the value associated with the last "w" is still 0.

(b) Correct on short sequences.

**ChatGPT**: It's very long so it might take a bit of time: 'w' for white, 'i' for ink, 'r' for read, 1 for yes/on/true and 0 for no/off/false. Set white paper. Ink: On. Ink: Off. Ink: On. Ink: Off....
And it goes on like this until you get to the 'r' at the end.
With the 'w' and 'r' commands at the beginning and end respectively, and 'i' commands between them, it's as if you are turning ink on and off multiple times on a white paper and then reading the results. However, without a pen to move on the paper, we can't get an actual image or text.
This sequence might make sense in the context of a robotic drawing or writing device that's being controlled to put down or lift up a pen to draw or write something on a paper, but without knowing the movement instructions, it's impossible to interpret what the final output should be.

(c) Wrong on long sequences (input length 1000).

Figure 8: Examples of interacting with ChatGPT-4 (as of 05/22/2023) by explaining FFLM to it.

example from Figure 2 (top right), and evaluate the capability of popular pretrained LLMs to correctly remember the state of a light switch. Specifically, `write` instructions in the FFLM task are either "Alice turns the light off" or "Alice turns the light on". The `ignore` instructions are either "Bob eats a banana" or "Bob eats an apple". All models are prompted with a translated, length-16 FFLM task that's been translated to English in this way before evaluation.

We measure this accuracy as a function of the sequence length for several well-known LLMs: GPT-2, GPT-2-large, GPT-2-xl, Pythia-12C, and GPT-NeoX-20B. Figure 3 shows how well these models perform on this task (i.e. the correctness of the model when prompted with "The light is turned ") as the sequence length is varied. Consistent with the findings of this paper, larger models tend to perform best at this task, and the quality of all models deteriorates with increased sequence length. Each point on the plot considers 500 sequences of the indicated length. All models were prompted with a randomly generated, length 16 flip flop sequence to allow the model to learn the task in context. Accuracy is measured according to the frequency with which the model correctly predicts the current state of the light switch, as described in Section B.2.1.

(R3) **10B-scale natural LMs can correctly process flip-flop languages, but not robustly.**

Note that it is impossible to quantify the degree to which these sequences are "in-distribution" (it is unlikely that any sequences of this form occur in the training distributions for these LLMs). Much like linguistic reasoning evaluations in the style of BIG-bench (Srivastava et al., 2022), we rely on the emergent capability of in-context inference (Brown et al., 2020) of the task's syntax and semantics. As discussed in Appendix A.4, this layer of indirection, which is impossible to avoid in the finetuning-free regime, can cause additional (and unrelated) failure modes to those studied in our synthetic experiments. Fully reconciling our findings between the synthetic and non-synthetic settings (e.g. by training or finetuning on sequences of this form, or via mechanistic interpretation of non-synthetic language models) is outside the scope of this paper, and yields an interesting direction for future work.

*Direct in-context induction of flip-flop languages?* Given the conversational abilities of LLMs, another way to interact with an existing pretrained model is to explain the definition of FFLM in natural language, and ask the model to output the correct state for `r`. We test this using ChatGPT (with GPT-4), as demonstrated in Figure 8. ChatGPT seems to understand the rules and can get short sequences correct (up to sequence length 400), but makes errors with unexpected connections on longer sequences.

## B.3    Effects of training data and scale (Section 5.1)

Here, we provide more details for the interventions outlined in Section 5.1, which directly modify the training distributions and resources.

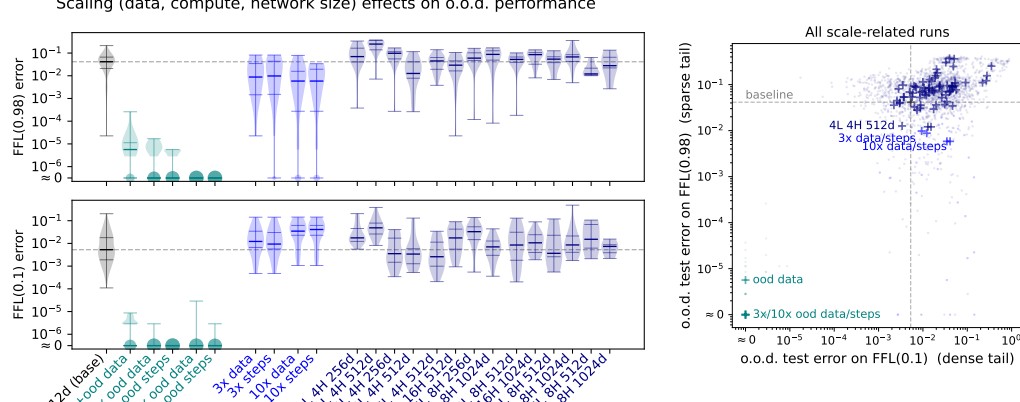

Figure 9: Detailed comparisons of various scaling axes, showing violin plots for selected runs *(left)* and an annotated scatter plot for all runs *(right)*; $+$ markers show medians for particular configurations. **Increasing training data diversity is by far the most effective way to mitigate attention glitches** in FFLM. The other scaling axes (increasing the amount of fresh data, increasing the number of optimization steps on the same dataset, and changing the model size) have mixed effects on rare-sequence performance. In particular, we wish to highlight that **neither overparameterization nor underparameterization strongly modulates the rate of glitches**.

**Long-tailed data.** For direct training on long-tailed sequences, we train on an equal mixture of $\mathsf{FFL}(0.8)$, $\mathsf{FFL}(0.98)$, and $\mathsf{FFL}(0.1)$, ensuring that the the sparse and dense tails are both adequately represented in training.

**Scaling up data and compute.** For both the default and long-tailed training distributions, we consider increasing the number of training iterations by a factor of $3$ or $10$, either with freshly generated samples ("$N\times$ data") or looping over the same training dataset for multiple epochs ("$N\times$ steps").

**Scaling up the model.** We also perform these exhaustive tail error evaluations on all 64 of the architecture scale settings shown in Figure 6, as well as a limited number of larger architectures (shown in Figure 9).

(R4) **Training on rare sequences works best, by a wide margin.** See the teal violins in Figure 9 *(left)*; training for longer (either with fresh data, or for more epochs on the same data) reduces the fraction of unlucky error-prone outliers. Recall that extrapolation is possible without such favorable coverage, via using a recurrent model.

(R5) **Resource scaling (in-distribution data, training steps, network size) helps.** Training on more data from the same distribution, as well as for more steps on the same examples, both improve sparse-sequence performance, at the expense of dense-sequence performance (blue violins in Figure 9 *(left)*). As best seen through Figure 9 *(right)*, there is no discernible monotonic relationship between any of the Transformer's standard architecture size parameters (i.e. number of layers, embedding dimension, and number of parallel self-attention heads per layer) and extrapolative performance (navy violins).

## B.4 Indirect algorithmic controls for extrapolation (Section 5.2)

As shown in Figure 4 in the main paper, various architectural, algorithmic and regularization choices can help improve error rates compared to the baseline Transformer. The various settings of weight decay, {attention, MLP, embedding} dropout, position embedding, activation function, attention-sharpening penalty are all found in Figures 10 and 11.

**Details for architecture variants.** There is no clear consensus on the advantages and drawbacks of various positional encodings, but it has been known Dai et al. (2019) that the choice of positional symmetry-breaking scheme modulates long-sequence performance on natural tasks. We evaluate

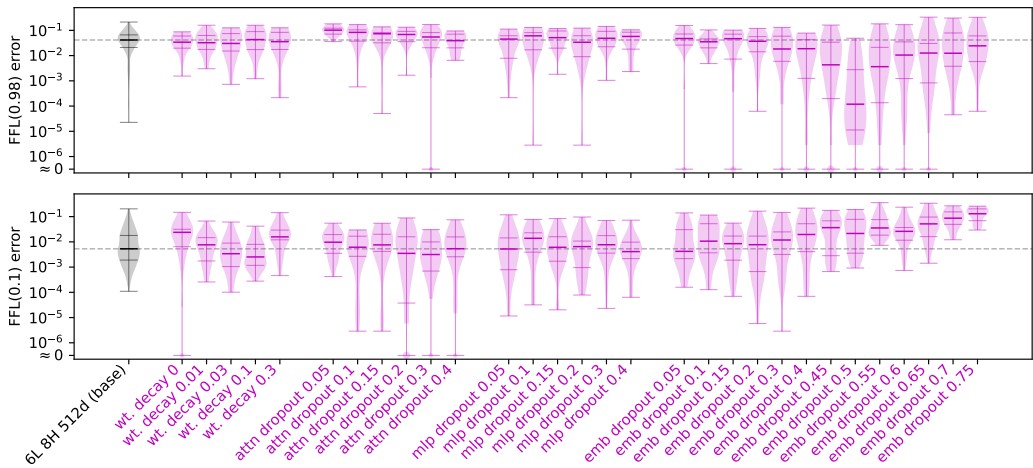

Figure 10: Detailed comparisons of standard regularizers (weight decay, and 3 forms of dropout). While some regularizer choices reduce rare-sequence error rates (in particular, large embedding dropout reduces sparse-sequence errors by 2 orders of magnitude), **nothing eliminates the glitches entirely**.

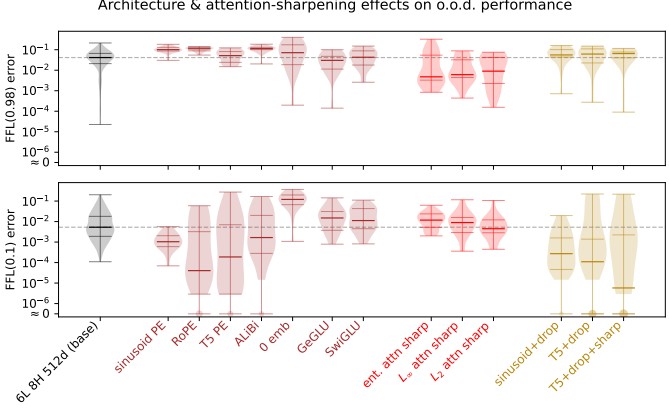

Figure 11: Detailed comparisons of a selection of architectural changes, attention-sharpening losses, and combinations of indirect algorithmic controls. **Our best runs come from jointly tuning these interventions**, including an annealing schedule for the attention-sharpening loss; however, **even the best models have nonzero glitch rates**. Figure 13 provides an exhaustive view of these results.

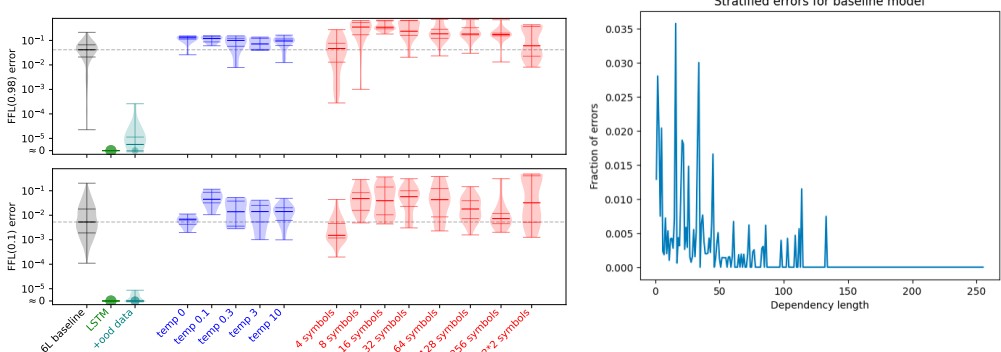

Figure 12: Supplementary plots. *Left:* Additional comparisons: modifying the softmax temperature $\beta$ inside the attention heads (multiplying the attention alignment scores by $1/\beta$), and generalizations of the FFLM task to larger vocabularies (i.e. the "data" tokens are uniformly drawn from $\{0, 1, \ldots, M-1\}$). In the rightmost violin, "2*2" refers to a token space consisting of length-2 bitstrings. In all of these settings, **attention glitches remain** (and mostly worsen). *Right:* Distribution of dependency lengths (number `ignore` operations since last `read` or `write` operation) of **wrong answers** on the union of the o.o.d. test sets, for a canonical baseline model. This serves as a quick check that **the errors are diverse**.

various choices which appear in high-profile LLMs: sinusoidal, learned, ALiBi (Press et al., 2021), and RoPE (Su et al., 2021). We also try the *zero* positional encoding (which breaks symmetry via the causal attention mask; see Haviv et al. (2022). We find that non-trainable position encodings help on dense sequences (FFL(0.1)), but have no clear benefit on sparse ones (FFL(0.98)) which require more handling of long-term dependency. We also try the gated activation units considered by (Shazeer, 2020).

**Details for attention sharpening.** There are many possible choices of continuous regularization terms which can promote sparsity in an attention head's weights—we consider entropy, negative $L_2$ loss, and negative $L_\infty$ loss. These terms are averaged across every attention head in the Transformer, and added as a surrogate objective during training. We perform a large grid sweep over coefficients $\{0.01, 0.03, ..., 0.1, 0.3, 1, 10, 30\}$, annealing schedules (linear and oscillating, starting from 0, 2000, and 5000 steps), and display in Figure 11 the 3 choices which appear on the Pareto front.

**Optimizer hyperparameters.** We also varied the optimizer parameters for AdamW ($\beta_1 \in \{0.85, 0.9, 0.95\}$, $\beta_2 \in \{0.95, 0.99, 0.999\}$, learning rate $\eta \in \{0.0001, 0.0003, 0.001, 0.003\}$) and found no significant improvements to extrapolation performance.

We restate the main findings:

(R6) **Many algorithmic choices influence extrapolative behavior.** We sweep over various forms of implicit and explicit regularizers; see Figures 10 and 11. Details are provided below.

(R7) **Despite many partial mitigations, nothing eliminates attention glitches entirely.** Refer to the scatter plot in Figure 4 (left) for a visualization of *every* training run.

Figure 12 provides a few supplementary experiments, verifying that attention glitches persist: scaling the attention softmax temperature, and larger vocabularies for the memory register. We also provide a quick verification that errors are not concentrated at the same dependency lengths (i.e. distances between `write` and `read`).

Finally, to provide a bird's-eye view of all of our experiments, Figure 13 provides a scatter plot of o.o.d. error rates for all models trained in this paper, color coded by intervention category.

## B.5 Preliminary mechanistic study and challenges (Section 5.3)

We continue the discussions in Section 5.3 and provide preliminary mechanistic interpretability results on *simulating the flip-flop automaton* (Definition 1). Recall the main takeaway:

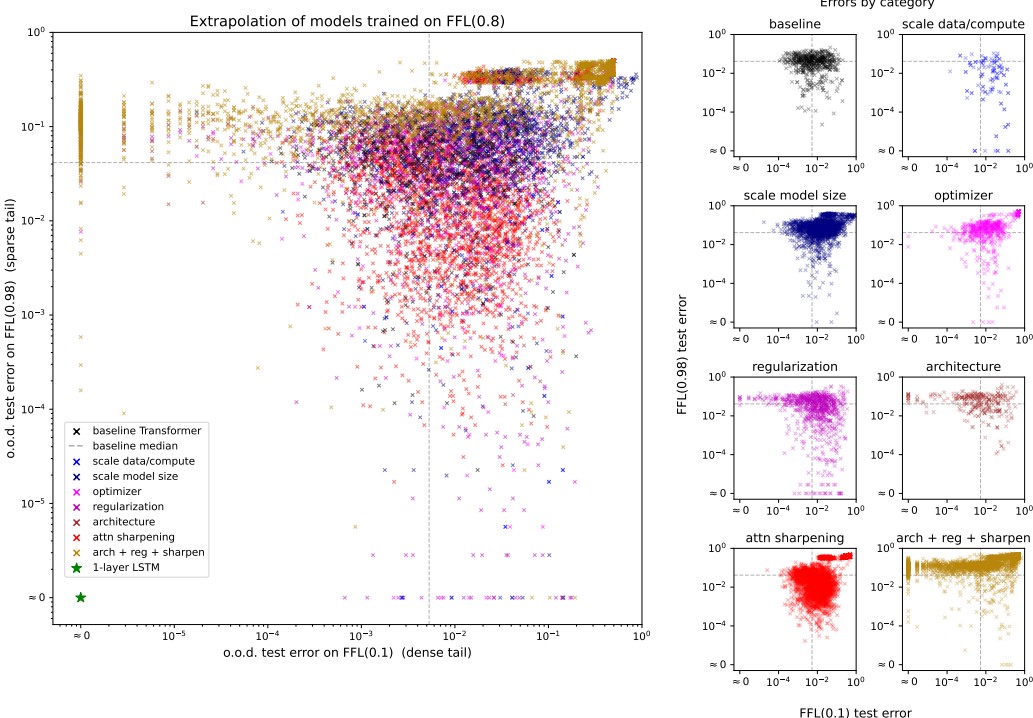

Figure 13: Error scatter plots for all models evaluated in this work (a larger version of Figure 4 *(left)* from the main paper), color-coded by intervention category. *Left:* All runs on the same plot. *Right:* Segmented by intervention type. The best runs result from an exhaustive joint hyperparameter search over dropout, position embeddings, and annealed attention-sharpening regularizers.

**(R8) Attention-sharpening regularizers successfully promote hard attention, but errors persist.**

**Sparsity regularization helps sharpen the attention.** Figures 16a and 16b compare the attention patterns of 1-layer 1-head models with or without attention-sharpening regularization. While both types of models give correct results, the attention-sharpened model puts all attention weights to the most recently `write` position, which is the solution given according to the definition of the task, whereas the attention patterns of the non-regularized model (Figure 16a) are much less clean.

*Are there solutions other than the "ideal" solution?* There is a solution naturally associated with the definition of the flip-flop automaton (i.e. the sparse pattern shown in Figure 16b), but it is not necessarily the *only* solution. For example, an equally valid (dense) solution is for the model to attend to every `write` token of the correct type.

*Are attention patterns reliable for interpretability?* Prior work has pointed out the limitations of interpretations based solely on attention patterns (Jain and Wallace, 2019; Bolukbasi et al., 2021; Wen et al., 2023). The intuition is that attention patterns can interact with other components of the network in various ways; for example, $W_V$ can project out certain dimensions even though they may have contributed to a large attention score. Hence, for multi-layer multi-head non-sparse models, the magnitude of attention weights may not have an intuitive interpretation of "importance" (Meister et al., 2021). This is observed in our experiments as well; for instance, Figure 17 shows examples where the attention on an incorrect token may be higher than that of the correct token. [17] However, in a 1-layer 1-head model, 1-sparse attention (Figure 16b) indeed offers interpretability, since if zero

---

[17]However, if we consider the "importance / influence" as measured by the norms of the attention-weighted value vectors, then the max norm still corresponds to the correct token, which helps explain why the final output is correct.

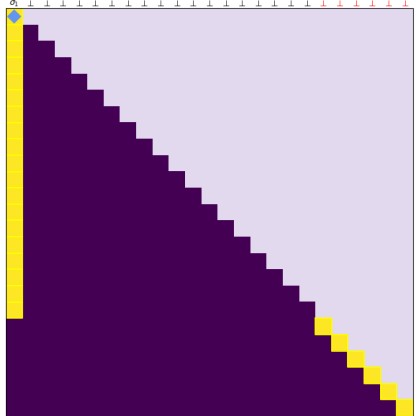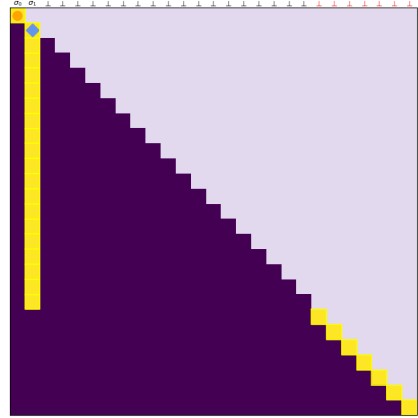

Figure 14: Attention drifts as the length increases. The model is trained on length-500 sequences with $p(\sigma \neq \perp) = 0.5$. The testing sequences are (a) $[2, \underbrace{0 \cdots, 0}_{800}]$, and (b) $[1, \underbrace{0 \cdots, 0}_{32}, 2, \underbrace{0 \cdots, 0}_{800}]$. We sample every 32 positions for visualization.

attention weight [18] necessarily means the absence of dependency, which greatly reduces the set of possible solutions implemented.

**Sporadic errors persist.**   Section Section 5.1 (R5) showed that none of the mitigations was successful at making Transformers reach 100% accuracy. One common failure mode is long-range dependency, where the input sequences contain very few `writes`. The failure could be attributed to multiple factors; we will explore one aspect related to attention patterns, demonstrated with a 1-layer 1-head Transformers with linear position encoding, on a length-834 sequence with 2 `writes`. As shown in Figure 14, the attentions for positions early in the sequence correctly attend to the most recent `write`. However, attention starts to "drift" as we move to later positions, and the positions at the end of the sequence attend entirely [19] to the recent `read` tokens, which contains no information for solving the task. This may be because the attention weights are undesirably affected by the position encodings, as discussed in Proposition 4.

**Optimization hurdles.**   While sparse solutions may be preferred for various reasons, sparsity itself is not sufficient to guarantee good performance: As shown in Figure 16d, sparsity regularization can lead to bad local minima, where the model tends to (incorrectly) rely on earlier positions. This is observed across different types of sparsity regularization. While we do not yet have a full explanation of the phenomenon, a possible explanation for this bias is that earlier positions show up more often during training, due to the use of the causal attention: a valid flip-flop solution is for the model to attend to every `write` token of the correct type; positions earlier in the sequence get observed in more subsequences because of the causal mask, and are hence more likely to be attended to. We also observe that the phenomenon seems to be closely related to the training distribution. For example, the model is much more likely to get stuck at a bad local minima when $p(\perp) = 0.5$ (denser sequences) compared to $p(\perp) = 0.9$ (sparse sequences). We leave a more thorough study on the training dynamics (Jelassi et al., 2022; Li et al., 2023; Ahn et al., 2023) for future work.

**Effect of sparsity regularization on training dynamics**   An interesting future direction is to understand the learning dynamics of flip-flop tasks with attention-sharpening regularization, as suggested by the (quantitively and qualitatively) different results and optimization challenges. As some initial empirical evidence that the regularization indeed have a large impact on the dynamics, we found that sharpened attention seems to have a regularization effect on the weight norms (Figure 15), and also lead to different behaviors of the attention heads (Figure 18).

---

[18]By "zero" we mean an attention score on the magnitude of 1e-8 empirically.

[19]The attention weights that are not on the most recent `write` sum up to around 1e-7.

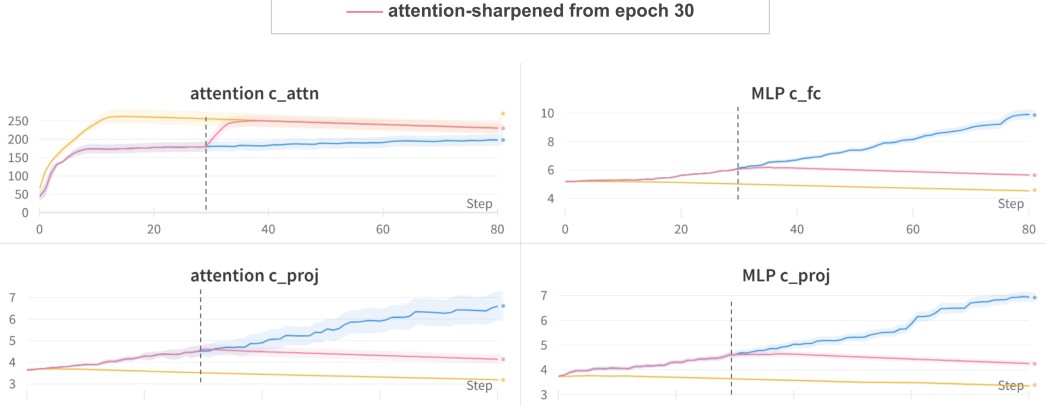

Figure 15: Frobenius norms of weight matrices in 1-layer 1-head models, trained without regularization (blue), with attention-sharpening regularization (yellow), or first without regularization and then adding regularization from epoch 30 (red; epoch 30 marked by the dashed lines). The solid curve and the shadow shows the median and the standard deviation calculated on 8 models.

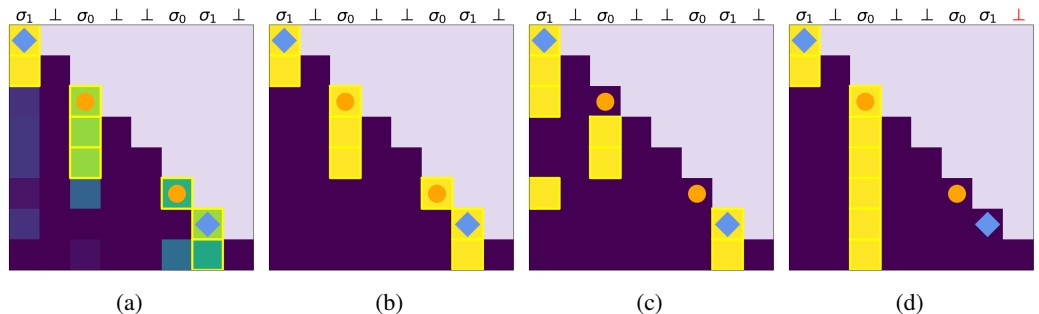

Figure 16: Attention-sharpening regularization on 1-layer 1-head models. Compared to a non-regularized model (16a), the sparsity-regularized model (16b) shows clear attention at the last write position. However, sparse attention does not have to align with the "ideal" pattern (16c), and can even be wrong (16d). Positions with yellow borders are where the max attention in each row occur; errors are marked in red.

**More examples of attention patterns** Figure 19 shows the full set of attention patterns of two 6-layer 8-head models trained with and without attention-sharpening regularization, corresponding to Figure 5 (a,b). Attention-sharpening regularization can be applied in different ways; for example, Figure 20 shows results of a model for which only the first layer is regularized. The attention patterns of subsequent layers remain sharp, even though there is no explicit regularization.

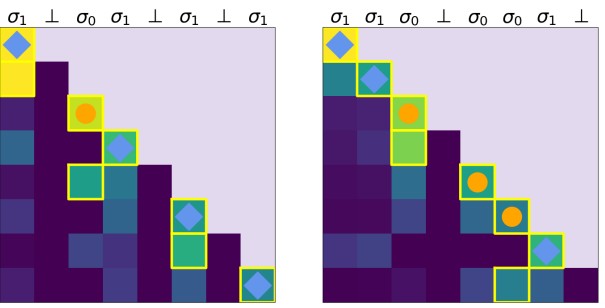

Figure 17: Non-sparse attention pattern can be misleading: a non-sparse model may put more attention on an incorrect token (i.e. a token that is not the `write` with the right type), while making the correct predictions. Yellow boxes mark the position of the max attention of each row.

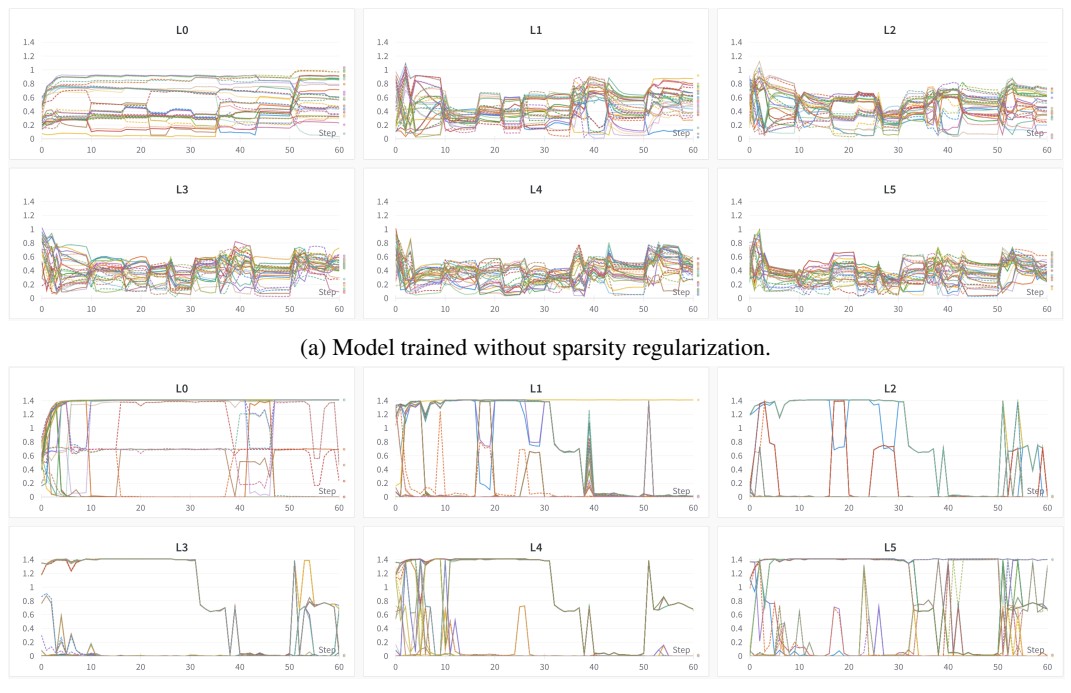

(a) Model trained without sparsity regularization.

(b) Model trained with entropy sparsity regularization with $\lambda = 0.01$.

Figure 18: Examples of the $\ell_2$ difference in attention patterns from two 6-layer 8-head 512-dimension models. Differences are calculated between all pairs of heads in the same layer.

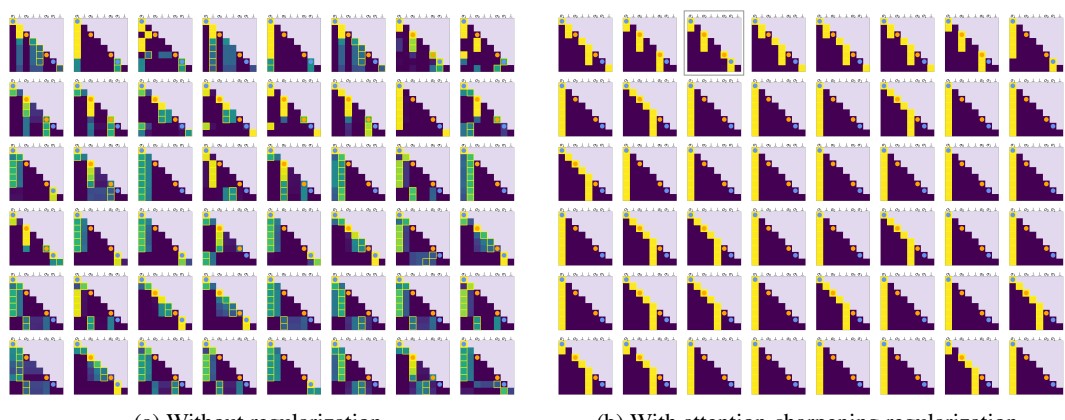

(a) Without regularization.          (b) With attention-sharpening regularization.

Figure 19: Attention patterns for 6-layer 8-head 512-dimension models on the input sequence $[\sigma_1, \perp, \sigma_0, \perp, \perp, \sigma_0, \sigma_1, \perp]$: attention-sharpening regularization lead to cleaner attention patterns. 1 attention head in the first layer of the regularized model (marked by the purple box) matches the "ideal" attention pattern Figure 5c.

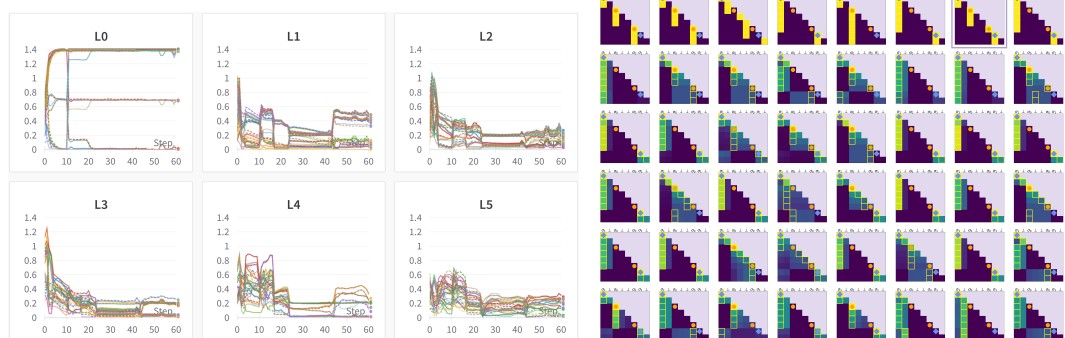

(a) $\ell_2$ differences between pairs of attention heads in the same layer, throughout training ($x$-axis).

(b) Attention patterns on the input sequence $[\sigma_1, \perp, \sigma_0, \perp, \perp, \sigma_0, \sigma_1, \perp]$.

Figure 20: Attention heads and attention patterns for a 6-layer 8-head 512-dimension model, trained with attention-sharpening regularization (entropy regularization with strength 0.01) on the first layer only. 1 attention head in the first layer (marked by the purple box) matches the "ideal" attention pattern Figure 5c.

### B.6 Software, compute infrastructure, and resource costs

GPU-accelerated training and evaluation pipelines were implemented in PyTorch Paszke et al. (2017). For the FFLM experiments, we used the `x-transformers`[20] implementations of the Transformer architecture and variants. For the fine-grained mechanistic interpretability experiments on the pure flip-flops, we used the "vanilla, GPT-2"-like Transformer implementation published by HuggingFace (Wolf et al., 2019). Our benchmarks have been released at `https://huggingface.co/datasets/synthseq/flipflop`.

Each training run was performed on one GPU in an internal cluster, with NVIDIA P40, P100, V100, and RTX A6000 GPUs, with at least 16GB of VRAM. Each (6-layer, 512-dimensional, 8-head) baseline model took ∼10 minutes to train (and evaluate online) for $10^4$ steps. A nontrivial fraction of the compute time ($\sim 20\%$) was spent on fine-grained evaluation through the course of training. The vast majority of training runs are close to these specifications; consequently, one set of replicates under identical conditions (i.e. each violin plot in each figure) is the product of ∼4 GPU-hours of training time.

We hope that this computational investment will aid in understanding how to build robust Transformer models and training pipelines at much larger scales.

## C Proofs for Section 4.1

**Transformer recap.** A Transformer (Vaswani et al., 2017) consists of multiple self-attention layers. Given $d$-dimensional embeddings of a length-$T$ sequence, denoted as $\boldsymbol{X} \in \mathbb{R}^{T \times d}$, a self-attention layer $f$ computes

$$f(\boldsymbol{X}) = \phi(\boldsymbol{W}_V \mathrm{softmax}(\boldsymbol{X}\boldsymbol{W}_Q \boldsymbol{W}_K^\top \boldsymbol{X}^\top)\boldsymbol{X}\boldsymbol{W}_V \boldsymbol{W}_C). \tag{C.1}$$

where $\boldsymbol{W}_Q, \boldsymbol{W}_K \in \mathbb{R}^{d \times k}$ for $k \leq d$ are the query and key matrix; $\boldsymbol{W}_V, \boldsymbol{W}_C^\top \in \mathbb{R}^{d \times k}$ project the representations from and back to $\mathbb{R}^d$. softmax calculates row-wise softmax. $\phi : \mathbb{R}^d \to \mathbb{R}^d$ is a 2-layer fully-connected network. Residual links and layer norm can be optionally included at different places of a self-attention layer.

### C.1 Realizability of FFL by small Transformers

**Proposition 2.** *A 2-layer 1-head Transformer with residual connections can represent "deterministic" FFL.*

---

[20]`https://github.com/lucidrains/x-transformers`

*Proof.* Let us consider predicting in the deterministic mode (Section 3.1). Then we need to predict $x_{t+1}$ given $x_{1:t}$ with $x_t = \mathtt{r}$. In order to do this, we need to find the largest $\tau < t$ such that $x_\tau = \mathtt{w}$ and output $x_{\tau+1}$. There are multiple ways to implement this, we will consider the following: (1) layer 1 converts FFL to the flip-flop automaton (Definition 1), (2) layer 2 implements the flip-flop construction. For layer 2, we can use the construction described in Liu et al. (2023). Here we present the full construction for completeness.

We will consider a two-layer Transformer with one head in each layer followed by a 2-layer MLP and a residual connection. In particular, for $x \in \{\mathtt{w}, \mathtt{r}, \mathtt{i}, 0, 1\}^T$:

$$f(x) = \phi_2(\boldsymbol{W}_V^{(2)} \mathrm{softmax}(f_1(x)\boldsymbol{W}_Q^{(2)}\boldsymbol{W}_K^{(2)\top}f_1(x)^\top)f_1(x)\boldsymbol{W}_V^{(2)}\boldsymbol{W}_C^{(2)})$$

$$\text{where } f_1(x) = E(x) + \phi_1(\boldsymbol{W}_V^{(1)}\mathrm{softmax}(E(x)\boldsymbol{W}_Q^{(1)}\boldsymbol{W}_K^{(1)\top}E(x)^\top)E(x)\boldsymbol{W}_V^{(1)}\boldsymbol{W}_C^{(1)})$$

where $E(x) \in \mathbb{R}^{T \times d}$ is the encoding for the input sequence $x$ given some encoding function $E$.

Our construction is as follows:

- Select $d = 7, k = 2, H = 1$ (recall from Equation C.1 that $d, k$ are the dimensions of $\boldsymbol{W}_Q, \boldsymbol{W}_K$). Among the $d = 7$ embedding dimension, two dimensions are for the operations ($\mathtt{w}$ versus $\mathtt{r}, \mathtt{i}$), two for the two $\mathtt{write}$ values, one for the positional embedding, one for padding, and the final dimension is for storing whether the previous position is the most recent $\mathtt{write}$, as calculated by the first layer.

- Select input symbol encodings such that for the token at position $t$, denoted as $x_t$,
$$E(x_t) := \mathbb{1}[x_t = \mathtt{w}]e_1 + \mathbb{1}[x_t = \mathtt{r} \vee x_t = \mathtt{i}]e_2 + \mathbb{1}[x_t = 0]e_3 + \mathbb{1}[x_t = 1]e_4 + e_5 + P_t \in \mathbb{R}^7,$$
where $P_t$ is the positional encoding. We use the linear positional encoding $P_t := (t/C) \cdot e_6$, for some (large) constant $C$. For a fixed sequence length $T$, we can set $C = T$.

- $\boldsymbol{W}_Q^{(1)} := \begin{bmatrix} e_5 & e_5 \end{bmatrix} \in \mathbb{R}^{7 \times 2}, \boldsymbol{W}_K^{(1)} := \begin{bmatrix} 3c\frac{e_1}{2T} & ce_6 \end{bmatrix} \in \mathbb{R}^{7 \times 2}$ for $c = O(T\log(T)), \boldsymbol{W}_V^{(1)} := \begin{bmatrix} e_1 & 0 \end{bmatrix} \in \mathbb{R}^{7 \times 2}$, and $\boldsymbol{W}_C^{(1)\top} := \begin{bmatrix} e_7 & 0 \end{bmatrix} \in \mathbb{R}^{7 \times 2}$.

- $\boldsymbol{W}_Q^{(2)} := \begin{bmatrix} e_5 & e_5 \end{bmatrix} \in \mathbb{R}^{7 \times 2}, \boldsymbol{W}_K^{(2)} := \begin{bmatrix} ce_7 & ce_6 \end{bmatrix} \in \mathbb{R}^{7 \times 2}$ for $c = O(T\log(T)), \boldsymbol{W}_V^{(2)} := \begin{bmatrix} e_4 & 0 \end{bmatrix} \in \mathbb{R}^{7 \times 2}$, and $\boldsymbol{W}_C^{(2\top} := \begin{bmatrix} e_1 & 0 \end{bmatrix} \in \mathbb{R}^{7 \times 2}$.

In layer 1, the unnormalized attention score for query position $i$ to key position $j$ is

$$\left\langle \boldsymbol{W}_Q^{(1)\top}x_i, \boldsymbol{W}_K^{(1)\top}x_j \right\rangle = \left\langle \frac{c}{T} \cdot \left[\frac{3}{2} \cdot \mathbb{1}[x_j = \mathtt{w}], j\right], [1,1]\right\rangle = \frac{c}{T} \cdot \left(\frac{3}{2}\mathbb{1}[x_j = \mathtt{w}] + j\right).$$

Note that the max attention value for position $i$ is achieved at $i$ if $x_{i-1} \neq \mathtt{w}$, else the max is achieved at position $i - 1$.

In the setting of hard attention, the output for the $i_{th}$ token after the attention module is $\mathbb{1}[x_{i-1} = \mathtt{w} \vee x_i = \mathtt{w}]e_7$. Now similar to the constructions in Liu et al. (2023) (Lemma 6), with a appropriate choice of $c = O(T\log T)$, we can approximate hard attention by soft attention, and subsequently use the MLP to round the coordinate corresponding to $e_7$. The MLP otherwise serves as the identity function. Together with the residual link, the first layer output (i.e. the second layer input) at position $i$ takes the form
$$f_1(x_i) = E(x_i) + \mathbb{1}[x_{i-1} = \mathtt{w} \vee x_i = \mathtt{w}]e_7.$$
In layer 2, the unnormalized attention score computed for position $i$ attending to $j$ is

$$\left\langle \boldsymbol{W}_Q^{(2)\top}f_1(x_i), \boldsymbol{W}_K^{(2)\top}f_1(x_j) \right\rangle = \frac{c}{T}\left\langle [1,1], \left[\mathbb{1}[x_{j-1} = \mathtt{w} \vee x_j = \mathtt{w}], \frac{j}{T}\right]\right\rangle$$

$$= c \cdot \left(\mathbb{1}[x_{j-1} = \mathtt{w} \vee x_j = \mathtt{w}] + \frac{j}{T}\right).$$

Note that the max attention value is achieved at the position right after the closest w to $x_i$. Let us denote this position by $\tau \leq i$, then with hard attention, the output at the $i_{th}$ position is $x_\tau e_1$, as desired. Now similar to before, we can approximate this with soft attention and use the MLP to do the appropriate rounding to get our final construction. $\square$

**Remark**: The construction in Proposition 2 is *a* construction, but it is not the *only* construction. For example, for the second layer implementation for the flip-flop automaton, there could be an equally valid *dense* solution, where the model uniformly attends to all `write` tokens of the correct type.

## C.2 Failure of soft attention: attention dilution with bounded Lipschitzness

Consider any attention layer with weight matrices $\boldsymbol{W}_Q, \boldsymbol{W}_K \in \mathbb{R}^{k \times d}$. If $\|\boldsymbol{W}_K^\top \boldsymbol{W}_Q\|_2$ is bounded, then the attention cannot be sparse as the sequence length increases:

**Proposition 3** (Leaky soft attention). *Assume the latent variables have bounded norm, i.e. $\|\boldsymbol{v}\|_2 \leq 1$ for any latent vector $\boldsymbol{v} \in \mathbb{R}^d$, and let $\sigma_{\max}$ denote the max singular value of $\boldsymbol{W}_K^\top \boldsymbol{W}_Q$. Then for $T = \Omega(\exp(2\sigma_{\max}))$, any sequences of latent vectors $\{\boldsymbol{v}_\tau\}_{\tau \in [T]}$, $\|\mathrm{softmax}(\{\boldsymbol{v}_\tau\}_{\tau \in [T]})\|_\infty = 1 - \Omega(1)$.*

*Proof.* The proof follows directly from a simple rewriting.

For any $\boldsymbol{u}, \boldsymbol{v}$ with $\|\boldsymbol{u}\|_2, \|\boldsymbol{v}\|_2 \leq 1$, the pre-softmax attention score is bounded by $\boldsymbol{u}^\top \boldsymbol{W}_K^\top \boldsymbol{W}_Q \boldsymbol{v} \in [-\sigma_{\max}, \sigma_{\max}]$.

$$\frac{\exp(\boldsymbol{v}_t^\top \boldsymbol{W}_K^\top \boldsymbol{W}_Q \boldsymbol{v}_T)}{\sum_{\tau \in [T]} \exp(\boldsymbol{v}_\tau^\top \boldsymbol{W}_K^\top \boldsymbol{W}_Q \boldsymbol{v}_T)} \leq \frac{\exp(\sigma_{\max})}{\exp(\sigma_{\max}) + (T-1)\exp(-\sigma_{\max})} = 1 - \frac{T-1}{T-1+\exp(2\sigma_{\max})},$$

where the last term is $\Omega(1)$ when $T = \Omega(\exp(2\sigma))$. $\square$

**Attention dilution and failure on dense sequences**   Strictly speaking, attention dilution caused by an increased sequence length does not necessarily affect the output of the layer. For example, if `ignore` gets mapped to a subspace orthogonal to that of `write`, then $\boldsymbol{W}_V$ can project out the `ignore` subspace, making the weighted averaged depending only on the number of `writes`. Hence with the presence of layer norm, attention dilution won't be a problem for the final prediction if the number of `write` is upper bounded regardless of the sequence length.

For the experiments in Section 5.1, denser sequences (i.e. larger $p(\texttt{write})$) do increase the number of `write` compared to the training distribution, hence attention dilution can be a potential cause for the decrease in performance.

## C.3 Failure of hard attention: bad margin for positional embeddings

In this section, we look at a failure mode that a 1-layer 1-head Transformer has on the flip-flop automaton simulation task. Why do we care about this setup? Simulating the automaton is in fact a sub-task of FFLM. For example, the second layer of the construction in Proposition 2 reduces to the simulation task.

Consider a 1-layer 1-head Transformer with parameters $\boldsymbol{W}_Q, \boldsymbol{W}_K \in \mathbb{R}^{k \times d}$. Write the attention query matrix $\boldsymbol{W}_Q$ as $\boldsymbol{W}_Q = [\boldsymbol{W}_{Qe}, \boldsymbol{W}_{Qp}]$, where $\boldsymbol{W}_{Qe} \in \mathbb{R}^{k \times (d-1)}$ corresponds to the embedding dimensions, and $\boldsymbol{W}_{Qp}\mathbb{R}^k$ corresponds to the dimension for the linear positional encoding. Write $\boldsymbol{W}_K = [\boldsymbol{W}_{Ke}, \boldsymbol{W}_{Kp}]$ similarly.

Then, we claim that the following must be true, regardless of the choice of the token embedding:

**Proposition 4.** *Consider linear positional encoding, i.e. $p_i = i/C$ for some (large) constant $C$. Then, perfect length generalization to arbitrary length requires $\boldsymbol{W}_{Qp}^\top \boldsymbol{W}_{Kp} = 0$.*

*Proof.* Let $\boldsymbol{e}^{(i)} \in \mathbb{R}^{d-1}$ denote the embedding vector (without the position encoding) for token $i \in \{0, 1, 2\}$. Let $\boldsymbol{v}_t = [\boldsymbol{e}_t, p_t]^\top \in \mathbb{R}^d$ denote the embedding for the $t_{th}$ token, where $\boldsymbol{e}_t \in \{\boldsymbol{e}^{(0)}, \boldsymbol{e}^{(1)}, \boldsymbol{e}^{(2)}\}\mathbb{R}^d$ is the embedding of the token itself, and $p_t := i/C$ is the linear positional encoding.

Let $s_{i \to j}$ denote the pre-softmax attention score that the $i_{th}$ token puts on the $j_{th}$ token, which is given by

$$s_{i \to j} = \langle \boldsymbol{W}_Q \boldsymbol{v}_i, \boldsymbol{W}_K \boldsymbol{v}_j \rangle \tag{C.2}$$

$$= \boldsymbol{e}_i^\top \boldsymbol{W}_{Qe} \boldsymbol{W}_{Ke} \boldsymbol{e}_j + \boldsymbol{e}_i^\top \boldsymbol{W}_{Qe}^\top \boldsymbol{W}_{Kp} \cdot p_j + (\boldsymbol{e}_j)^\top \boldsymbol{W}_{Ke} \boldsymbol{W}_{Qp} \cdot p_i + \boldsymbol{W}_{Qp}^\top \boldsymbol{W}_{Kp} \cdot p_i p_j \tag{C.3}$$

$$= \boldsymbol{e}_i^\top \boldsymbol{W}_{Qe} \boldsymbol{W}_{Ke} \boldsymbol{e}_j + \frac{\boldsymbol{e}_i^\top \boldsymbol{W}_{Qe}^\top \boldsymbol{W}_{Kp}}{C} \cdot j + \frac{(\boldsymbol{e}_j)^\top \boldsymbol{W}_{Ke} \boldsymbol{W}_{Qp}}{C} \cdot i + \frac{\boldsymbol{W}_{Qp}^\top \boldsymbol{W}_{Kp}}{C^2} \cdot ij. \tag{C.4}$$

We will prove the proposition in two cases, which respectively require $\boldsymbol{W}_{Qp}^\top \boldsymbol{W}_{Kp} \le 0$ and $\boldsymbol{W}_{Qp}^\top \boldsymbol{W}_{Kp} \ge 0$.

**Case 1: $\boldsymbol{W}_{Qp}^\top \boldsymbol{W}_{Kp} \le 0$ required**    Consider the case of long-term dependency, where the input sequence consists of an initial write and a series of reads, i.e. $\sigma_1 = 1$ and $\sigma_t = 0$ for $t > 1$. Then for the $T_{th}$ position, the score for the first write token is

$$s_{T \to 1} = \langle \boldsymbol{W}_Q \boldsymbol{v}_T, \boldsymbol{W}_K \boldsymbol{v}_1 \rangle \tag{C.5}$$

$$= \boldsymbol{e}^{(0)\top} \boldsymbol{W}_{Qe} \boldsymbol{W}_{Ke} \boldsymbol{e}^{(1)} + \frac{\boldsymbol{e}^{(0)\top} \boldsymbol{W}_{Qe}^\top \boldsymbol{W}_{Kp}}{C} + \frac{(\boldsymbol{e}^{(1)})^\top \boldsymbol{W}_{Ke} \boldsymbol{W}_{Qp}}{C} \cdot T + \frac{\boldsymbol{W}_{Qp}^\top \boldsymbol{W}_{Kp}}{C^2} \cdot T \tag{C.6}$$

$$= \left( \frac{(\boldsymbol{e}^{(1)})^\top \boldsymbol{W}_{Ke} \boldsymbol{W}_{Qp}}{C} + \frac{\boldsymbol{W}_{Qp}^\top \boldsymbol{W}_{Kp}}{C^2} \right) \cdot T + O(1) = O(T), \tag{C.7}$$

and the score for the last write token is

$$s_{T \to T} = \langle \boldsymbol{W}_Q \boldsymbol{v}_T, \boldsymbol{W}_K \boldsymbol{v}_T \rangle \tag{C.8}$$

$$= \boldsymbol{e}^{(0)\top} \boldsymbol{W}_{Qe} \boldsymbol{W}_{Ke} \boldsymbol{e}^{(0)} + \frac{\boldsymbol{e}^{(0)\top} \boldsymbol{W}_{Qe}^\top \boldsymbol{W}_{Kp}}{C} T + \frac{\boldsymbol{e}^{(0)\top} \boldsymbol{W}_{Ke} \boldsymbol{W}_{Qp}}{C} \cdot T + \frac{\boldsymbol{W}_{Qp}^\top \boldsymbol{W}_{Kp}}{C^2} \cdot T^2 \tag{C.9}$$

$$= \frac{\boldsymbol{W}_{Qp}^\top \boldsymbol{W}_{Kp}}{C^2} \cdot T^2 + O(T). \tag{C.10}$$

Think of $C$ as going to infinity. If $\boldsymbol{W}_{Qp}^\top \boldsymbol{W}_{Kp} > 0$, then there exists a sufficiently large $T$ such that $s_{T \to T} > s_{T \to 1}$. Hence we need $\boldsymbol{W}_{Qp}^\top \boldsymbol{W}_{Kp} \le 0$.

**Case 2: $\boldsymbol{W}_{Qp}^\top \boldsymbol{W}_{Kp} \ge 0$ required**    Consider the input sequence where $\sigma_1 = 1$, $\sigma_{T-1} = 2$, and $\sigma_t = 0$ for $t \in [T] \setminus \{1, T-1\}$. Similar to the above, calculate the pre-softmax attention scores for $\sigma_1, \sigma_{T-1}$ as

$$s_{T \to 1} = O(T) \tag{C.11}$$

$$s_{T \to T-1} = \frac{\boldsymbol{W}_{Qp}^\top \boldsymbol{W}_{Kp}}{C^2} \cdot T^2 + O(T). \tag{C.12}$$

Since we need $s_{T \to T-1} > s_{T \to 1}$, it must be that $\boldsymbol{W}_{Qp}^\top \boldsymbol{W}_{Kp} \ge 0$.

$\square$

