# OpenReview forum: "Exposing Attention Glitches with Flip-Flop Language Modeling"
_NeurIPS.cc/2023/Conference — NeurIPS 2023 spotlight_

### Official Review · Reviewer_4Fu7 · 2023-06-29

**Soundness:** 4 excellent
**Presentation:** 4 excellent
**Contribution:** 4 excellent
**Rating:** 9
**Confidence:** 4

**Summary:**

The paper introduces Flip-Flop Language Modeling (FFLM) to examine closed-domain
hallucinations of Large Language Models (LLMs). Flip-Flop languages are
synthetic benchmarks which model a single bit of memory and its operations: read
(r), ignore (i), and write (w). For example, the string "w 1 i 0 r 1" is valid
as the same bit was written and read, but "w 1 i 0 r 0" is invalid. The task of
the Transformer is to attend to the last write operation and generate the
correct bit for the read operation.

The authors discover that the Transformer exhibits a long tail of reasoning
errors even on this simple task (they call these attention glitches), and these
errors are very hard to mitigate. Interestingly, the LSTM can perform the task
perfectly, with 20x less training.

They also introduce a preliminary study on the effect of attention sharpening on
training Transformers to simulate the flip-flop automaton. Intuitively
sharpening should help as the Transformer only needs to attend to the most
recent write. They find that sharpening can indeed help but also introduce new
errors.



**Strengths:**

The paper provides very interesting and significant insights into the workings
of the Transformer architecture.

Some of the most interesting findings:
- Transfomers make reasoning errors on the long tail
- 1-layer LSTMs can learn the same task perfectly
- Transformers exhibit high variance in the OOD test error between random seeds
  and even iterations withing the same training run
- 10B-scale LLMs cannot solve the task robustly
- architectural and regularization changes can have orders-the-magnitude effects
  on OOD performance
- in contrast to LSTMs, it seems that completely eliminating attention glitches
  is not possible for Transformers

Flip-flop languages are also interesting as they are realizable by
self-attention (as the authors explain in lines 221-224), so they could
represent a reasoning task which should actually be easier for LLMs.

The paper is also very well written, I could not find any errors or opaque
sections.



**Weaknesses:**

I think the main weakness of the study is that it's limited to flip-flop
languages. That being said, I think this is just because of the foundational
nature of the study: examining other languages would be outside the scope of the
paper and could be explored in future work.

Still, although the findings are very interesting, some of them may be specific
to flip-flop languages. For example, (R4), training on rare sequences, would be
very hard to do in the general case of natural language, and so I'm not sure
what (R4) means in the general case. Also, (R5) says that scaling improvements
are orders of magnitude smaller, I wonder if that stands for natural language
too as it's much more diverse.

The authors include some natural language experiments isomorphic to a flip-flop
language and a discussion about natural language in the Appendix.

Even though Section 5.3 is a preliminary study it would be good to include some
quantitaive results.

I believe that Fig. 5 (d) is incorrect: the error marked in red should be
$\sigma_0$ and $\sigma_1$.

The notation $\Delta$ could be introduced, I don't think it's widely
used in probability.



**Questions:**

In the flip-flop language the value can be only 0 or 1. It could be interesting
to extend the task to more values (like 0-15, or a single byte, 0-255). How
worse do you think the Transformer and the natural LMs would perform in that
case? Do you think that the LSTM would still perform very well?

The attention sharpening regularization uses $-\Vert\alpha\Vert_{2}$, but usually
$L_1$ norm is used for sparsification. I found the reasoning behind
$L_\infty$ and entropy in the cited paper (Zhang et al.), but I still don't
see how using $L_2$ norm sparsifies. Could you elaborate on this?



**Limitations:**

The limitations are addressed thoroughly in the Conclusion.

---

> ### Author Rebuttal · Authors · 2023-08-10
>
> Thank you very much for the encouraging feedback and for the helpful suggestions! We are glad that you find our results interesting and significant, and would like to discuss the remaining concerns below.
>
> **[W1] Generalizing insights to natural languages**
>
> This is a great question; we provided some discussions in [G1] of the global response. We’d also like to comment on “incorporating rare sequences'', which is as of now the only effective solution. While it is unclear how to formally define “rare sequences” in natural languages, we intuitively think of it as “improving data diversity”, which has been proven effective by several related works in various setups. The closest to our setting is Jelassi et al. 2023, which showed that adding as few as 10 OOD samples can greatly improve the generalization performance on arithmetic tasks. More generally, data diversity is shown to be the main contributor of CLIP’s robustness (Fang et al. 2022), and a properly chosen data mixture can improve the pretraining results (Nguyen et al. 2022, Xie et al. 2023).
> Regarding scaling improvement, we have some partial evidence that while scaling helps improve the performance, it cannot fully eliminate the errors. Please see (R3) in our paper, as well as discussions in Appendix B.2.
>
> **[W2] Quantitative results on mechanistic interpretability**
>
> In Appendix B.5, we provided quantitative results on the regularization’s effect on weight norms (Figure 11) and difference in attention patterns (Figure 14, Figure 16(a)). We are happy to incorporate further suggestions to the results!
>
> **[W3] Potential typo in Fig 5(d)**
>
> Thank you for the close read! The figure is actually correct, that is, the error is indeed on the read token marked as the red $\bot$. Note that a seemingly incorrect attention pattern does not necessarily mean a wrong prediction; please see our footnote 8 and the discussion in “Appendix B.5 – Sparsity regularization helps sharpen the attention – Are attention patterns reliable for interpretability?” In this case, Even though the attention for the last $\sigma_0$ and $\sigma_1$ are on the wrong token, their predictions are still correct. We apologize for the confusion and will make this clear in the revision.
>
> **[Q1] - FFLM with multiple symbols**
>
> We experimented with more symbols and report the results in Figure 2 of the global response PDF. Note that the 1-layer LSTM solves all of the task variants perfectly; please see [G4] in our global response for more discussions.
>
> **[Q2] Attention sharpening with L2**
>
> Note that we are regularizing on vectors after the softmax, where each attention vector has an L1 norm equal to 1 (i.e. on the simplex, which is denoted by $\Delta$). The largest L2 norm on the simplex is achieved when the vector is one-hot, hence encouraging a large L2 norm is equivalent to sharpening the attention. We will clarify this in the revised paper, and also define $\Delta$ clearly.

---

> > ### Comment · Reviewer_4Fu7 · 2023-08-13
> > **Thank you for your response**
> >
> > Thank you for your insightful comments and for pointing out the quantitaive
> > results in the Appendix. I also appreciate Figure 2 in the response. I believe
> > that the paper will be even better with the proposed revisions.
> >
> > I still strongly believe that this negative result is very significant, the
> > paper should be accepted, and it could be the basis for much future work. To
> > emphasize this I increased my score to 9.

---

> > > ### Author Response · Authors · 2023-08-21
> > >
> > > We are pleased that our response addressed your concerns, and thank you very much again for your valuable suggestions and support!

---

### Official Review · Reviewer_ycg1 · 2023-07-03

**Soundness:** 4 excellent
**Presentation:** 4 excellent
**Contribution:** 3 good
**Rating:** 8
**Confidence:** 4

**Summary:**

The paper identifies a simple task for which the Transformer architecture fails. The authors introduce flip-flop language modeling, towards quantifying the extrapolation capabilities of different architectures. Transformer models are determined to suffer from long tail errors, in a phenomenon termed "attention glitches".

Throughout the paper, this new synthetic task is analysed and the failure modes of Transformer models are identified. The authors embark and compare different techniques to mitigate the failures. Some theoretical justification and some mechanistic interpretability are also provided.

**Strengths:**

1. The paper discusses a very simple task for which recurrent networks triumph, but Transformer present failure cases, attributing these failures to the inductive bias of the model. The task proposed is very simple and adequately motivated.
2. Long range dependencies are a common failure case of Transformer models. FFLM allows for easy to check failures for reasoning tasks.
3. Extensive experiments with different regularizations are provided.

**Weaknesses:**

1. I feel some of the implications and connections to Ji et al., 2023 are not highlighted. From Figure 3, depth does not seem to have a direct consequence on performance.
2. Although some theoretical justification is provided, assumptions are in cases too strong, i.e. linear positional encoding.
3. Although a series of regularization and other techniques are proposed to solve the errors, no clear solution is proposed that works.

**Questions:**

1. It will be helpful to report errors as a function of difficulty. This can be done by presenting the errors as a function of sequence length (as done in Figure 3-right) or by also presenting errors as a function of different FFL tasks. If we define as $d$, the distance between the last write and the current read operation, is there a connection between the distribution of $d$ in the training set and the types of errors in the extrapolation set? How many examples of distance $d$ need to be presented for the model to generalize? For (possible) outliers $d$ in the training set, is the model fitting them correctly? There might be some interesting connections with Grokking [1, 2], where the model switches from a state of memorizing to one of generalizing.
2. Preposition 2 states that a 2-layer 1-head Transformer can represent FFL. Is there is practice any benefit with depth?
3. Other ways to decrease the entropy of the attention include the change of the temperature inside the softmax (which is not included in Proposition 3), or the method of [3].

Fixes:
Figure 3 (left) x-axis is cropped (50 -> 500), also middle-right (10->10K).

[1] Power, Alethea, et al. "Grokking: Generalization beyond overfitting on small algorithmic datasets." arXiv preprint arXiv:2201.02177 (2022).

[2] Thilak, Vimal, et al. "The slingshot mechanism: An empirical study of adaptive optimizers and the grokking phenomenon." arXiv preprint arXiv:2206.04817 (2022).

[3] Martins, André, et al. "Sparse and continuous attention mechanisms." Advances in Neural Information Processing Systems 33 (2020): 20989-21001.

**Limitations:**

The paper proposes a simple example to study failure in extrapolation of Transformer models. The authors identify the lack of correct inductive bias in the model as the major shortcoming. Although many different regularization techniques are proposed, no clear answer is given for this phenomenon. I believe that this will however steer the community towards the right direction, for answering fundamental questions what these models is concerned.

---

> ### Author Rebuttal · Authors · 2023-08-10
>
> Thank you very much for the careful read and for the insightful comments! We are glad that you find the FFLM task to be well-motivated and helpful for steering the direction of the community towards fundamental concerns about the models. We hope to address your concerns below.
>
> **[W1] Connection to hallucination and insights from Ji et al. 2023**
>
> This is a great question. Please refer to [G1] in our global response.
>
> **[W2] Some assumptions for theory are strong**
>
> As our experiments highlight, the OOD distribution errors have a long-tail behavior and the OOD performance itself varies drastically across different random seeds, as well as, across different iterates in the same runs. This makes it highly non-trivial to develop a complete end-to-end theory for attention glitches. Our work takes the first step towards this by characterizing the impact of the self-attention head on the failure of OOD. We understand that the assumption on linear position encoding and focusing on a single attention-head is strong, but we believe the insights here are already useful in highlighting the challenge of FFLM and how simple fixes may not be enough to fix the OOD problem.
>
> **[W3] No clear mitigation strategy is proposed.**
>
> Please see [G3] in our global response.
>
>
> **[Q1] Error as a function of “difficulty”; relation to grokking**
>
> We have added an additional plot (please see Additional Figure 3 in the rebuttal pdf) highlighting the fraction of errors by dependency length. Note that the errors occur at both short and long dependency range. Plotting fine-grained OOD performance using richer metrics of the training set beyond the ignore token probability $p$ used to generate it is a great suggestion. Note that $p$ itself does characterize the distribution of distance between read and write for the training set (as long as we have enough samples), which is why we used this simple metric. We would be happy to add additional experiments looking more closely at the relationship of errors and training distribution. We agree that understanding how much of the OOD data of each type is necessary to generalize well is a very interesting question for future work.
>
> Moreover, our results are not directly related to grokking, since the training is done in an online fashion (i.e. each batch of samples is freshly sampled) and hence there is no overfitting.
>
>
> **[Q2] Practical benefit / Role of depth of Transformer**
>
> Proposition 2 emphasizes that depth-2 is sufficient to represent the FFLM language. We believe depth-2 is necessary as well. In our experiments, we not only train Transformers with depth-$2$ but also depth-$\{4, 6, 8\}$. (We also provide a few runs at larger depths, up to $16$; see Figure 1 in the rebuttal attachment.) However, we do not see a significant correlation between depth and OOD performance in the $>10000$ training runs with various hyperparameters. Since depth adds more representation power, it might seem that it offers an overparameterization benefit. However, it also introduces entanglements with optimization dynamics, where the benefits/drawbacks of depth are not so clear. There has been some recent work [1] that highlights that bigger models might end up being less robust than smaller models. We believe that understanding the role of overparameterization would be an interesting direction for future work.
>
>
> **[Q3] Attention sharpening by changing the temperature**
>
> Following [2], we used a $\log(\tau t)$ temperature, where $\tau$ is the temperature parameter, and $t$ is the position (i.e. we use a different temperature for each position). However, this method was not able to help mitigate OOD errors, as shown in Figure 2 of the global response PDF.
>
> **References**
>
> [1] Miceli-Barone, A. V., Barez, F., Konstas, I., and Cohen, S. B. (2023). The larger they are, the harder they fail: Language models do not recognize identifier swaps in python. arXiv preprint arXiv: 2305.15507.
>
> [2] David Chiang, Peter Cholak. Overcoming a Theoretical Limitation of Self-Attention.

---

> > ### Comment · Reviewer_ycg1 · 2023-08-15
> >
> > Thank you for the additional results, I believe they are very valuable.
> >
> > The authors have address all my concerns. If I understand the "Additional Figure 3" correctly, the authors present the location of errors as a function of distance to the last read. As smaller distances are a lot more likely, perhaps it would make more sense to present the probability of error prediction as a function of that distance, i.e. how many errors were made for distance $d$, compared to how many instances of distance $d$ can be found.
> >
> > Overall, I find this work and the FFLM fascinating and valuable to future research, as a simple yet challenging toy experiment. I believe authors have addressed concerns from all reviewers. To reflect this I have increased my score.

---

> > > ### Author Response · Authors · 2023-08-16
> > >
> > > Thank you for the follow-up and the encouraging feedback!
> > >
> > > **Clarification regarding the "errors vs. distance" plot:** Additional Figure 3 *does* normalize by the count (hence the label "fraction of errors"). You are correct that the distribution of denominators is extremely lopsided, due to the geometric distribution of dependency lengths; hence, the former plot (plotting counts instead of fractions) would be uninformative. (At the upper range, there may be undetected errors, due to rarity of such dependencies even under the FFL(0.98) distributions.) We'll clarify this upon merging into the manuscript.

---

### Official Review · Reviewer_vbMc · 2023-07-04

**Soundness:** 2 fair
**Presentation:** 2 fair
**Contribution:** 3 good
**Rating:** 6
**Confidence:** 3

**Summary:**

This work studies the phenomenon of "attention glitches" in LLMs. Attention glitches are instances where an LLM's attention mechanism fails to capture long-range dependencies, resulting in factual inaccuracies or erroneous reasoning. The authors introduce a new synthetic benchmark called flip-flop language modeling (FFLM) to probe the extrapolative behavior of LLMs. FFLM is a simple generative task that requires an LLM to copy binary symbols over long-range dependencies, ignoring the tokens in between. The authors find that Transformer Transformer-based FFLMs suffer from a long tail of sporadic reasoning errors, even when the task is relatively simple.

**Strengths:**

- The paper studies a critical problem on attention glitches for transformer-based models. It can potentially have a huge impact on the community.
- The synthetic FFLM benchmark can be useful to study long-range dependency in a controllable way.
- The findings/research questions provide more insights on the issue.

**Weaknesses:**

- The paper discussed and aimed at drawing a connection between hallucination and long-range dependency, however, there is no further discussion on how these two terms interact with each other.
- While it is understandable that the paper studied the behavior of attention glitches, there is no proposed path to fix the issue which limits the contribution of this work.
- The setting is fully synthetic. It is unclear how any findings can be transferred into realistic and more complex tasks.
- Some experimental settings that are critical to the conclusion are unclear, e.g., dataset construction and sequence lengths, tokenizer choice, etc.

**Questions:**

- The paper showed that 1-layer LSTM works perfectly for FF while transformers are not. However, it is unclear to me whether that is due to an effect of overfitting (to the distribution of training data). Could authors elaborate more on this?
- "We hypothesize that attention glitches occur in the internal algorithmic representations of Transformer models of natural language, and that they account for (a non-negligible portion of) the reasoning errors encountered in practice. " Is there a way to validate the hypothesis?
- It would be great if the authors can release the code to reproduce the results.

**Limitations:**

- It should be made clear that the full experiments and analysis are based on synthetic setting.

---

> ### Author Rebuttal · Authors · 2023-08-10
>
> Thank you very much for your careful review! We are glad that you find the attention glitches phenomenon to be a critical problem that is worth studying, and hope to address your concerns below.
>
> For the first three concerns in Weakness, please refer to the global response:
> - **[W1] Hallucination vs long-term dependency?** Please see [G1] in our global response.
> - **[W2] Is there a clear fix?** Please refer to [G3] in our global response.
> - **[W3] how do insights transfer to more complex / real-world setups?** Please refer to [G2] in our global response.
>
> Below we would like to discuss the remaining questions.
>
> **[W4] Details about experimental setups**
>
> Attention glitches are found even as we vary the dataset, sequence length, and tokenization. Figure 3 (right) demonstrates that attention glitches become more frequent as sequence length increases, and that this phenomenon appears across multiple commonly studied LLMs. Many of these models have different tokenization strategies and training sets. We have provided experimental details in Appendix B; please let us know if there are particular setups that you are concerned about, and we are happy to clarify further.
>
> **(Q1) Is Transformer failing because of overfitting?**
>
> We interpret “overfitting to training distribution” as “overfitting the empirical distribution of training samples” (please let us know if we misinterpreted your question), which we do not think is the reason. Note that we train both LSTM and Transformers on online data where each batch is freshly sampled (as described in the detailed setups, i.e. Appendix B.2 – Training and evaluation data), which can be considered as training for 1 epoch and hence unlikely to overfit.
>
> **(Q2) Connecting attention glitches to hallucination in natural languages?**
>
> Please refer to [G2] in our global response.
>
> **(Q3) Code release**
>
> We plan to release the code upon publication, thank you for the suggestion!
>
> **(L1) Clarify all experiments are synthetic**
>
> We believe this is rather clear already, since we have explicitly stated this in multiple places:
> - FFLM is stated as “a parametric family of synthetic benchmarks…” in the abstract (line 9), in the list of contributions at the end of the introduction (line 47), and in the conclusion section (line 292).
> - Section 3.2 also clearly states that FFLM is synthetic and is dedicated toward justifying the importance of the benchmark.
> - The beginning of Section 4 (line 162) states that the experiments are synthetic.
>
> Please let us know if you have suggestions on better highlighting the setup, thank you very much!

---

> > ### Comment · Area_Chair_XhKr · 2023-08-18
> >
> > Thank you for the detailed response and for addressing the reviewers' points.
> >
> > We have pinged reviewer vbMc, and they will make sure to go over the rebuttal soon.
> >
> > Your AC

---

> > ### Comment · Reviewer_vbMc · 2023-08-20
> >
> > I'd like to thanks the authors for providing a very detailed explanation and I apologize for the late response. The responses particularly the general one helped clarify the paper a lot, and consequently I have raised my score to 6. Please do incorporate these points in the final paper.

---

> > > ### Author Response · Authors · 2023-08-21
> > >
> > > Thank you very much again for the time and effort you've dedicated to reviewing our paper! We really appreciate your suggestions and will make sure that they are reflected in the camera ready version.

---

### Official Review · Reviewer_FpxV · 2023-07-05

**Soundness:** 3 good
**Presentation:** 3 good
**Contribution:** 3 good
**Rating:** 6
**Confidence:** 4

**Summary:**

This paper introduces Flip-Flop Language Modeling (FFLM), a synthetic benchmark designed to evaluate language model's (LMs) ability to perform operations on a single-bit of memory. LMs are evaluated on their ability to generalize to out-of-distribution (OOD) sequences. The training setups are varied along several axes including hyperparameters (random seed, training steps, weight decay, dropout), architectural changes (number of layers, hard vs. soft attention, positional encodings), and dataset properties (additional OOD data). In all cases, the LMs produce attention glitches, i.e., they exhibit errors in their reasoning. In contrast, an LSTM generalizes perfectly. The authors provide preliminary hypotheses explaining the failure modes.

**Strengths:**

*Originality:* although synthetic benchmarks have been often studied in NLP, FFLM focuses on the smallest reasoning capability (memory on a single bit), reducing confounding factors driving mistakes.

*Quality:* the experiments are comprehensive and cover most of the "standard" training and architectural tricks. Moreover, the FFLM benchmark is well-designed and is appropriately justified as a benchmark. These two factors lead to a useful analysis of current limitations of LMs.

*Clarity:* the paper is well-written and figures + captions are self-explanatory

*Significance:* understanding reasoning errors in LMs is a crucial step towards improving their reliability and thus future deployment in real-world systems. Moreover, this work provides a minimal benchmark with which to evaluate any proposed architectural changes to LMs.

**Weaknesses:**

The main weakness with the work is a lack of actionable insights. Although several mechanistic hypotheses are proposed as to explaining the behavior behind the benchmark, they are not adequately explored nor clearly organized. This leads to the following concerns:

**Evaluation concerns:**

* *Lack of experiments studying whether the proposed hypotheses in Section 4.1 actually drive the errors examined.* I appreciate the compilation of possible hypotheses explaining the LM errors. However, there don't appear to be experiments attempting to falsify these hypotheses? For example, one could examine whether the LMs do indeed only attend to the previous $n$-tokens.

**Clarity concerns:**
* *Does not emphasize the benchmark properties of FFLM.* Although the authors defined success on FFLM as achieving 100% performance accuracy, if people are to evaluate their architectural innovations on FFLM, it would be helpful to have a more continuous notion of progress (rather than solved vs. unsolved). Possible suggestions include:
    * Defining a notion of difficultly of generalization. E.g., FFL(0.1) is harder than FFL(0.2) given training data from FFL(0.98). Perhaps one could introduce an easy and hard test set.
    * Classifying severity of failures. Is there perhaps some metric that would estimate how close the LM attention patterns match either the ideal solution or any other possible solution? Intuitively, there should be some distinction between the attention weights of a randomly-initialized network and the attention weights of a trained model with error (Figure 5d). Perhaps one metric could be the proportion of attention weight placed on all the previously encountered write tokens vs. on all other tokens

* *Should have a clearer list of future directions.* I think it would help guide future work if there was a compiled list of actionable experiments and / or hypotheses to test that were provided in the appendix.

**Questions:**

Questions:
1. Were there any other hypotheses explored in Section 4.1? It would be nice to have a more careful enumeration (perhaps by category) of the possible hypotheses driving the lack of OOD generalization. Moreover, one of the proposed mechanisms is already present in previous work.

Typos:
1. Line 867: "the regularization indeed have" --> "the regularization indeed has"

**Limitations:**

Although there appears to be a limitations section in Appendix A.4, I think it would be appropriate to more clearly note in the main body that this work does not posit a full list of possible mechanisms driving poor performance on FFLM nor will solving performance on FFLM constitute a transformer with robust reasoning.

---

> ### Author Rebuttal · Authors · 2023-08-10
>
> Thank you very much for your thoughtful comments! We are glad that you found findings from our minimally sufficient FFLM task to be original and significant. There were some great points raised in the review, which we’d like to address below.
>
> **Providing a list of actionable future directions**:
>
> Thank you for the great suggestion! Here are our current thoughts:
> - *Learning from synthetic setups*: Our results show that simple synthetic setups can serve as clean, controllable sandboxes for isolating and analyzing various mechanisms in the architecture. Designing benchmarks or probing tasks like FFLM could help better understand Transformer’s capabilities and limitations.
> - *FFLM extensions and connecting to natural languages*: An obvious limitation of a synthetic setup is its gap to the real-world settings. One direction is to study whether and how findings can transfer to real-world setups; as an example, please see our response [A1] to Review 4Fu7 regarding the effect of data. Another direction is to expand the synthetic setup to narrow the gap to natural languages, for which we provided several directions in Section A.4, such as incorporating more complex selection criteria and expanded notion of ground truth.
> - *More error analyses and (mechanistic) understanding*: While our paper provides explanations of two failure modes and some preliminary mechanistic study, a more comprehensive understanding of Transformer’s internal mechanism can be helpful. In contrast to single-component analyses in our work, such understanding under natural languages setups can be much more challenging and nuanced; for example, please see our discussion “Submodules” in Section A.4.
> - *Better metrics*: as the reviewer pointed out, a continuous notion of progress can be much more informative than a binary answer or accuracy. One direction is to take inspiration from works on hidden progress throughout training, such as Barak et al. 22, Nanda et al. 23.
> - *Architectural innovations*: Attention glitches is essentially a problem with the architecture: Transformer fails at FFLM, while LSTM succeeds easily. Therefore, architectural innovations are likely a necessary part of the fix. As discussed in Section 6, one idea is to integrate the recurrent inductive bias into the Transformer architecture, which several prior works have shown promising results [Katharopoulos et al. 2020, Dao et al. 2022, Peng et al. 2023].
>
>
> **Better understanding the task and the failures**
>
> As the reviewer correctly noted, the success metric of our proposed task is to process FFLM perfectly, which can be practically thought of as performing as well as recurrent networks (LSTMs). A continuous progress measure could be the accuracy, and it is a great question to find other metrics that could serve as more informative progress measures; please see the list of future directions above. Comparing attention weights is an interesting idea that is worth trying, though our current guess would be that it might not be directly helpful, since various prior work has shown that attention patterns can be misleading as discussed in Footnote 7 and Appendix B.5.
>
> To better highlight the effects of various mitigation methods on the errors, we added Figure 1 in the global response PDF, which shows that each type of methods may be helpful for improving the OOD error on either the denser sequences or the sparser sequences, but no method was able to improve both simultaneously.
>
>
> **Hypothesis driving lack of OOD performance**
>
> We explored several potential hypotheses that lead to poor OOD performance (not explicitly organized in Section 4.1). Let us reiterate them here organized by the different aspects of the training pipeline:
> - (Architecture) _H1: Small Transformers are not capable of representing the FFLM language perfectly_. We refuted this by showing a theoretical construction of a 2-layer (low parameter count & low-norm) Transformer that realizes FFLM with 100% accuracy.
> - (Architecture) _H2: Soft attention can lead to dilution_. We supported this hypothesis by showing that attention sharpening indeed improves generalization across sparse sequences.
> - (Architecture) _H3: Transformers can only focus on the recent $n$ tokens_: We show that this explanation is insufficient since the model makes errors even on dense sequences.
> - (Data) _H4: OOD generalization requires coverage of tails_. We support this by showing that adding sparse and dense examples improves OOD performance.
> - (Optimization) _H5: Bad local minima exist_. We support this in our interpretability studies by showing that even after enforcing hard attention, the optimization can find solutions that confidently choose the wrong argmax on certain sequences.
>
> Note that none of the hypotheses fully explains all the failure modes we observe in our experimentation, which helps highlight the challenge of explaining OOD failures using simple mechanisms. Towards developing an end-to-end explanation, we would need to incorporate the interplay between different attention heads in each layer and across different layers, and the impact of optimization on the in-distribution loss on the OOD loss, which is highly non-trivial.
>
> Thank you also for the suggestions on clarifying the limitations of setup and for correcting the typo; we will update the camera ready version accordingly.

---

> > ### Comment · Reviewer_FpxV · 2023-08-13
> > **Thank you for your details response.**
> >
> > I appreciate the proposed list of directions and hope they will be included in the paper. I would like to see this paper accepted but I am keeping my score.

---

> > > ### Author Response · Authors · 2023-08-21
> > >
> > > We will make sure to include these discussion in the camera ready version. Thank you very much again for your support and for the valuable suggestions!

---

### Official Review · Reviewer_ZZvs · 2023-07-06

**Soundness:** 3 good
**Presentation:** 4 excellent
**Contribution:** 2 fair
**Rating:** 6
**Confidence:** 3

**Summary:**

This paper presents a paradigm for diagnosing one cause of closed-domain (intrinsic) hallucinations in large language models (LLMs) and presents its analysis results.

To understand why LLMs can be factually inaccurate or prone to erroneous reasoning, the authors proposed a flip-flop language model (FFLM) task designed to verify behavior during extrapolation. By controlling the probability of ignoring behavior, the FFLM can generate sequences with both dense-and-short and sparse-and-long dependencies.
This allows us to observe changes ofTransformer's learning outcomes and behavior between these sequences.

The analysis revealed that occasional, long-tail inference failures cause errors, and currently, their occurrence cannot be predicted. Some of the effects can be mitigated with regularization, but they cannot be completely eliminated.

The main claims are as follows:

- Proposing the flip-flop language modeling task. This task is designed to probe issues during the inference of long sequences easily.
- Transformers can learn flip-flop language, but regardless of dependency length, they occasionally cause long-tail errors.
- To mitigate this attention glitch, several regularization techniques are verified. No method can resolve the problem entirely, though each technique has a certain effect.

### after rebuttal
I increased my score from 5 to 6 based on the explanations provided in the feedback.

**Strengths:**

- Proposal of FFLM, a controllable long-range dependency resolution task, under a simple task setting. The adjustability of the dependency range by the ignore action's probability is easy to understand.
- According to the authors, this is the first attempt to attribute model hallucinations to a systematic architectural flaw in the Transformers.
- Presenting various experimental results and possible explanations from many aspects. Unfortunately, none in this report are definitive causes and/or solutions (most are denied by exceptions), but they can be utilized as one of the initial investigation items in the analysis of natural language LM's hallucination.
- It's interesting that LSTMs and 1-layer single Transformers can solve some issues, but multi-layer LSTMs are not good at the issues. This suggests the necessity of architectural innovations beyond just stacking self-attention layers.

**Weaknesses:**

- While the paper presents several interesting results to the readers, currently, the manuscript does not provide a clear guideline on what steps should be taken to counteract hallucinations in natural language LLMs. The impression I get from the current manuscript is that 'our current knowledge does not pinpoint the cause of hallucinations in natural language LLMs'."

- I found it interesting that LSTMs and 1-layer single Transformers sometimes performs better than the multi-layer LSTM. Why does this happen? I believe there must be important implications, so if I have overlooked the explanation, please point it out in the review answers.


**Questions:**


This is just a simple idea: Consider a unit with more memory bits. Would it be possible to create a unit that defines the ignore probability for each bit independently?
Is it possible that a multi-head attention can learn FFLM (with more bits) efficiently, by aligning the number of bits and heads?

**Limitations:**

Well discussed in the manuscript.

---

> ### Author Rebuttal · Authors · 2023-08-10
>
> Thank you very much for the thoughtful comments and great questions! We are glad that the reviewer finds it interesting that FFLM exposes the failure of multi-layer Transformers when smaller architectures can perform better, and hope to address the concerns below.
>
> **Why LSTM / smaller Transformers perform better**
>
> There might be some typos in the reviewer’s original question, and we would like to make sure our understanding is correct: we interpret the question as: “why can 1-layer LSTM and 2-layer Transformers perform better than multi-layer Transformers?”
> Representationally, both 1-layer LSTM and 2-layer Transformers are minimally sufficient to solve the task. Comparing these two, we hypothesize that the LSTM has a recurrent inductive bias that is more suitable for the FFLM task, which is consistent with prior works aiming to solve some reasoning tasks via 'recurrent prompting' [1,2,3] or by introducing recurrence into the architecture directly [4,5,6].
> Comparing 2-layer and multi-layer Transformers, the latter performs worse likely due to the redundancy in the architecture, which allows more solutions that are consistent with the training samples but can have more unexpected behaviors on unseen samples.
>
> **This is a comprehensive negative result paper**
>
> While Section 5 documents a wide selection of algorithmic interventions that can quell the attention glitch pathology by orders of magnitude, it is true that we do not provide a solution on par with data diversity or the recurrent network. The goal of this paper is to highlight one specific kind of hallucination, and to emphasize that this is particular to the Transformer, regardless of its scale or specific architecture; recurrent predecessors have no such shortcoming.
>
> FFLM offers a precise probing mechanism for benchmarking progress on hallucinations of this kind, and further, helps to identify one concrete, disambiguated cause of the general LLM hallucination problem. We believe that advancing language models requires both thorough experimentation that identifies specific, classifiable kinds of hallucinations and honed tools like FFLM to study them closely — our work makes these contributions.
>
> **Generalizations beyond 1-bit memory.**
>
> There are several ways to generalize the FFLM, as we discussed in Appendix A.4. Below are some options most related to generalizing the memory:
> - Keeping the instruction set to be the same, and using a larger set of values. For example, two isomorphic ways to write a sequence could be “$w\ 3\ i\ 2\ i\ 1\ r\ 3$” (“4 symbols”) and “$w\ 1\ 1\ i\ 1\ 0\ i\ 0\ 1\ r\ 1\ 1$” (“2*2 symbols”). Figure 2 in the global rebuttal attachment shows that errors persist (and, in fact, both long- and short-range error rates worsen) with increasing vocabulary size.
> - Keeping the set of values to be the same, and using a larger instruction set: for example, when there are 2 memory units (indexed as 0 and 1), a sequence could be “$w_0\ 0 \ w_1\ 1\ r_1\ 1\ r_0\ 0$” where $w_i$ ($r_i$) refers to write at (read from) memory bit $i$. This has more flexibility in the probability hyperparameters and can be solved using a construction similar to the one given in Proposition 2, but extended to using more heads. This is analogous to interleaving multiple FFLM tasks and slightly tangential to our current investigation, and we are happy to include more results in the camera-ready.
>
> **References**
>
> [1] Maxwell Nye et al. Show Your Work: Scratchpads for Intermediate Computation with Language Models.
>
> [2] Jason Wei, Xuezhi Wang, Dale Schuurmans, Maarten Bosma, Brian Ichter, Fei Xia, Ed Chi, Quoc Le, Denny Zhou. Chain-of-Thought Prompting Elicits Reasoning in Large Language Models.
>
> [3] Bingbin Liu, Jordan T. Ash, Surbhi Goel, Akshay Krishnamurthy, Cyril Zhang. Transformers Learn Shortcuts to Automata.
>
> [4] Angelos Katharopoulos, Apoorv Vyas, Nikolaos Pappas, François Fleuret. Transformers are RNNs: Fast Autoregressive Transformers with Linear Attention.
>
> [5] Daniel Y. Fu, Tri Dao, Khaled K. Saab, Armin W. Thomas, Atri Rudra, Christopher Ré. Hungry Hungry Hippos: Towards Language Modeling with State Space Models.
>
> [6] Bo Peng et al. RWKV: Reinventing RNNs for the Transformer Era.

---

> > ### Comment · Reviewer_ZZvs · 2023-08-16
> > **Thank you for the clarifications**
> >
> > Authors,
> >
> > thank you for your feedback! (and excuse me for typos...)
> >
> > Concerning "Why LSTM / smaller Transformers perform better": Yes, that is what I want to ask. The provided explanation sounds convincing to me. Thanks!
> >
> > I find the answers to my questions and fellow reviewers' questions are overall convincing and satisfactory.
> > I'm leaning towards a more positive attitude, so I will increase my score to 6.

---

> > > ### Author Response · Authors · 2023-08-21
> > >
> > > We are glad that our response addressed your concerns. Once again, thank you very much for the insightful discussions and support!

---

### Author Rebuttal · Authors · 2023-08-10

We thank all the reviewers for their thoughtful comments. In this global response, we address questions posed by multiple reviewers, and outline additional experiments we ran during the author response period.

**[G1] Gap between FFLM and natural languages:** We certainly agree that there are numerous discrepancies between FFLM vs. natural language modeling. We introduce FFLM not to act as a representative distribution for the many diverse capabilities needed to process natural language; rather, we use it to isolate *one* important capability where Transformers exhibit subtle errors. We reiterate the perspectives discussed in our paper:
* Instances of FFLM are embedded in distributions of natural language and code, as seen in Figure 2c. We see in practice that natural LLMs exhibit similar sporadic errors when prompted to complete such sequences (Figures 1 and 3 (top right)). Thus, robust FFLM processing is a necessary (but far from sufficient) condition for robust language processing.
* Shallow, parallel compositions of flip-flops can process a large class of formal languages known to be relevant in syntactic parsing and algorithmic reasoning [Liu et al. ‘23]. Thus, FFLM is an “atomic” unit of more complex sequence processing capabilities, and attention glitches may provide a way to tackle the notoriously hard problem of diagnosing the internal representations of LLMs. We are excited to tackle this in future work.

**[G2] Gap between attention glitches and hallucinations in practice:**
* We are only claiming that Transformers’ errors on flip-flop strings to be similar to a specific type of hallucination: namely, closed-domain hallucinations, where the model’s generations contradict unambiguously presented factual information provided in the context. Our intent is to provide a _minimal_ example which fulfills this criterion and reveals a shortcoming in the inductive bias of Transformers.
* We _do not_ claim to address (or even define) LLM hallucinations in their full scope; the full question of defining LLM factuality is ambiguous and philosophical in nature.
* As noted by reviewer ycg1, work by [Ji et al., 2023] suggests that LLM hallucinations broadly tend to improve with increasing depth, but here we find that deeper language models are not necessarily more resilient to attention glitches. Attention glitches are a particular kind of hallucination that Transformer architectures are susceptible to, and the architectural particulars of a Transformer-based model (like depth) do not remedy them. Our work, which considers various architectures, tokenizers, and regularizers, suggests that more invasive interventions are necessary to solve attention glitches.

**[G3] Regarding algorithmic fixes:** Reviewers ZZvs and ycg1 shared concerns about lack of algorithmic fixes. Rather than a weakness, we view this to be our work’s **central negative result**: we searched comprehensively for a clear fix, and did not find one (apart from changing the distribution or eschewing the Transformer).

While Section 5 documents a wide selection of algorithmic interventions that can quell the attention glitch pathology by orders of magnitude, it is true that we do not provide a solution on par with data diversity or the recurrent network. The goal of this paper is to highlight one kind of hallucination, and to emphasize that this is particular to the Transformer, regardless of its scale or specific architecture; recurrent predecessors have no such shortcoming. FFLM offers a precise probing mechanism for benchmarking progress on hallucinations of this kind, and further, helps to identify one concrete, disambiguated cause of the general LLM hallucination problem. We believe that advancing language models requires both thorough experimentation that identifies specific, classifiable kinds of hallucinations and honed tools like FFLM to study them closely — our work makes these contributions.

**[G4] Additional experiments and plots:** During this response period, we launched a few more sets of training runs, to address some of the reviewers’ curiosities. Figures are provided in the attachment.
* Dependence on scale: Figure 1 shows violin plots for o.o.d. errors across various model sizes, showing no clear trend (and no significant gain from increasing or decreasing model size). This corresponds to the hyperparameter grid sweep outlined in Figure 6 in the appendix of the original manuscript.
* Temperature: Prior work has suggested that sparsity of attention heads can be achieved by scaling the attention scores before the softmax by a constant (i.e. tuning the temperature). Our preliminary finding is that this does not improve extrapolation (Figure 2); note that direct attention sharpening *does* work (though not perfectly).
* Different number of states: Reviewers ZZvs and 4Fu7 both suggest variants of this experiment (which is also mentioned as a natural generalization axis in the paper); see our response to ZZvs for an explanation of the variants. Our preliminary findings: the long-tail errors persist; the 1-layer LSTM solves all of the task variants perfectly.
* Stratified errors by dependency length: in Figure 3, as requested, we exhibit a breakdown of error rates as a function of dependency length (distance from the current read token to the previous non-ignore token), on 3000 sequences from FFL(0.1) and 30K sequences from FFL(0.98)). This shows, in a finer-grained manner, that the errors are diverse in nature (not concentrated on any particular $n$-gram length), corroborating that this phenomenon evades oversimplified characterizations.

---

### Decision · Program_Chairs · 2023-09-21

**Decision:**

Accept (spotlight)

**Comment:**

This paper proposes a synthetic task, called Flip-Flop Language Modeling (FFLM), for evaluating "attention glitches" in language models (LMs), which are cases when the model fails to capture long-range dependencies, resulting in factual inaccuracies or erroneous reasoning. The task simulates read/write/ignore operations over a single bit of memory, that requires copying binary symbols over long-range dependencies while ignoring the tokens in between. The authors find that transformers struggle on this simple task, with errors that are very hard to mitigate, while LSTMs can perform the task perfectly with 20x less training.

The main concerns raised by the reviewers and discussed during the rebuttal focused on (a) lack of actionable insights, (b) the fact that the paper discusses a connection between hallucinations and long-range dependencies, without discussing how these two terms interact with each other, and (c) the evaluation being limited to a fully synthetic task, making it unclear if the findings will generalize to natural languages and more complex tasks. Most of these concerns were adequately addressed by the authors, who listed concrete ways for future work to use their findings, and clarified the scope of their work.

Overall, there was a general agreement that the paper makes substantial contributions, both introducing a succinct and valuable evaluation for LMs and exposing a key limitation of the core neural architecture that has been dominating the NLP field (and is widely used in other fields as well). Findings from this work might be valuable for tackling the problems of long-range dependencies and hallucinations in language modeling, which are central problems today.